# Interactive Person Retrieval via Multi-Turn Multimodal Conversation

Yang Bai [1 2]  Tingfeng Wang [1]  Bin Yang [1]  Min Cao [3]  Jinqiao Wang [2 4]  Mang Ye [1]

## Abstract

Traditional text-based person retrieval approaches typically rely on single-shot textual queries, which are generally incomplete or vague in real-world scenarios. Recently, chat-based person retrieval methods enable iterative query refinement via question-answering interactions between the system and users. However, these methods fall short of direct user interaction with retrieved candidates during conversation, making it challenging to effectively refine the retrieval results. To address these limitations, we propose multimodal interactive person retrieval (MInterPR), a new retrieval paradigm that allows users to iteratively refine retrieved candidates by providing feedback on visual differences from the target person. To support this task, we establish MInterPEDES, a multimodal conversational dataset constructed by augmenting existing question-answering dialogues with synthesized visual feedbacks. Furthermore, to tackle the challenge of accurate and efficient semantics understanding in multimodal dialogues, we propose a multimodal conversational memory-enhanced framework, MNEMO, which incorporates an atomic turn encoding (ATE) module to model each dialogue turn independently, and a dialogue memory aggregation (DMA) module to capture the fine-grained information and cross-turn dependencies. Extensive experiments demonstrate that MNEMO achieves substantial improvements in both retrieval accuracy and generalization ability, highlighting its promising potential in real-world scenarios. The code and dataset are publicly available at https://github.com/Flame-Chasers/MNEMO.

---

[1]National Engineering Research Center for Multimedia Software, School of Computer Science, Wuhan University, Wuhan, China [2]Wuhan AI Research, Wuhan, China [3]School of Computer Science and Technology, Soochow University, Suzhou, China [4]Institute of Automation, Chinese Academy of Sciences, Beijing, China. Correspondence to: Mang Ye <yemang@whu.edu.cn>.

*Proceedings of the 43rd International Conference on Machine Learning*, Seoul, South Korea. PMLR 306, 2026. Copyright 2026 by the author(s).

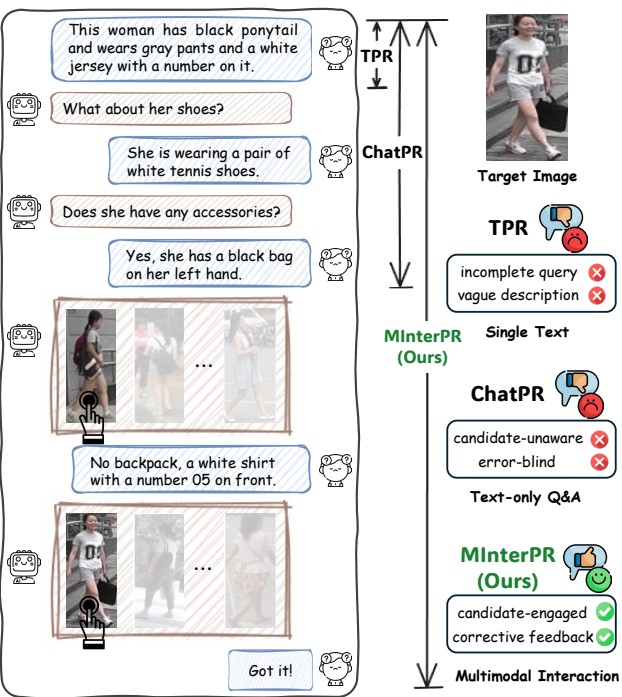

*Figure 1.* Comparison of queries in text-based person retrieval (TPR), chat-based person retrieval (ChatPR), and our multimodal interactive person retrieval (MInterPR). TPR relies on a single text without interaction. ChatPR adopts textual question-answering dialogues, lacking direct user interaction with retrieval candidates. MInterPR supports multi-turn multimodal dialogues, enabling users to inspect candidates and provide corrective feedbacks, progressively refining the retrieval results towards the target person.

## 1. Introduction

Person retrieval has become increasingly critical in the fields of public safety and intelligent surveillance. In this context, text-based person retrieval (TPR) (Niu et al., 2024; Irene et al., 2024) employs free-form natural language queries, effectively addressing the practical limitation of traditional visual queries, known as person re-identification (Ye et al., 2022; 2024; Yao et al., 2025; Hong et al., 2025; Leng et al., 2020), where visual examples of the queried person may not always be readily available in real-world scenarios. Consequently, TPR has gained ever-rising attention in recent years (Jiang & Ye, 2023; Qin et al., 2024; Zuo et al., 2025).

Existing methods in TPR typically rely on single-shot text queries to describe the target individual, which are often

incomplete or vague in real-world scenarios, leading to sub-optimal retrieval performance. Recently, chat-based person retrieval (ChatPR) (Bai et al., 2025; Lu et al., 2025; Xie et al., 2025; Qin et al., 2025) introduces an iterative query refinement process, allowing users to interact with the system via question-answering. However, ChatPR fails to enable users to directly engage with retrieved candidates, leading to a disconnect between user feedback and retrieval results, and making it difficult to pinpoint errors and refine the retrieval process. To bridge this gap, we propose multimodal interactive person retrieval (**MInterPR**), a novel task that enables users to provide corrective feedbacks based on differences between retrieved candidates and the target person during the conversation. As illustrated in Figure 1, both TPR and ChatPR can be regarded as special cases of MInterPR, substantially broadening its applicability and generalization in practical person retrieval scenarios.

However, when tackling MInterPR, two critical challenges arise: **(1) Lack of multimodal conversational data**: Although recent years have witnessed an increase in the availability of TPR datasets (Li et al., 2017; Ding et al., 2021; Zhu et al., 2021; Zuo et al., 2024b), there remains a lack of multimodal conversational data for person retrieval. This scarcity significantly impedes progress in exploring and benchmarking MInterPR methods. **(2) Challenge in accurate and efficient compositional semantic understanding**: In the context of MInterPR, models are required to accurately and efficiently understand complex compositional semantics from multimodal conversations. However, existing methods that independently encode each modality (Jiang & Ye, 2023; Cao et al., 2024) struggle to capture the compositional semantics inherent in multimodal conversational feedbacks. Meanwhile, dialogue modeling methods (Bai et al., 2025; Lu et al., 2025; Levy et al., 2023; Zhao et al., 2025) that concatenate all dialogue turns into a single long context not only fail to effectively extract fine-grained person details from complex interaction, but also suffer from increased computational overhead as interactions grow, which presents a significant barrier for real-world person retrieval systems. Therefore, a key challenge is to design models that can effectively integrate multimodal conversational cues with both accuracy and efficiency.

In response to the first challenge, we introduce **MInter-PEDES**, a multi-turn multimodal dataset for person retrieval, where the multimodal conversational data are constructed by extending existing question-answering dialogues (Bai et al., 2025) with synthetic user feedbacks. For each target image, we first sample a set of visually similar reference images to simulate retrieval candidates during conversation. Given the prohibitive cost of large-scale human annotation, we employ multimodal large language models (MLLMs) (OpenAI, 2024; Chen et al., 2024; Han et al., 2023) to automatically generate difference descriptions between the refer-

ence and target images, mimicking user-provided corrective feedbacks. These feedbacks, along with the corresponding reference images, are integrated with existing textual dialogues as additional turns to form multimodal conversations. Finally, we perform meticulous manual curation on the constructed data to rectify inappropriate cases, such as generation errors and incomplete feedback, thereby ensuring the high quality and reliability of our dataset.

To address the second challenge, we propose a multimodal conversational memory-enhanced framework (**MNEMO**), which adopts a hierarchical architecture comprising an Atomic Turn Encoding (ATE) module and a Dialogue Memory Aggregation (DMA) module. ATE independently models the multimodal dialogue semantics within each conversational turn, effectively circumventing the computational inefficiency of lengthy contexts, which could induce quadratic complexity for transformer-based architectures. On the other hand, DMA aggregates and contextualizes the turn-level semantics through two types of memory queries: static queries to capture common semantic patterns and fine-grained information, and dynamic queries to propagate conversational dependencies throughout the entire dialogue. With the synergy of ATE and DMA, the proposed MNEMO framework achieves efficient and accurate semantic understanding in multimodal conversational contexts, thereby enabling more precise person retrieval.

***Contributions:***

- We introduce a new person retrieval paradigm, multimodal interactive person retrieval (MInterPR), which enables users to iteratively refine the retrieval results through corrective feedbacks, progressively steering the system towards the target person image during the conversation.

- We construct and release the first multi-turn multimodal conversational dataset for person retrieval, MInterPEDES, providing the community with a foundational and available benchmark to advance MInterPR research.

- We propose a multimodal conversational memory-enhanced framework (MNEMO), consisting of an atomic turn encoding module to independently encode each dialogue turn, and a dialogue memory aggregation module to capture cross-turn dependencies.

- Extensive experiments demonstrate the effectiveness of MNEMO in handling multi-turn multimodal conversations, highlighting its promising potential in real-world person retrieval scenarios.

## 2. Related Work

### 2.1. Text-based Person Retrieval

Text-based person retrieval aims to locate specific person images based on free-form textual descriptions. Early stud-

ies (Li et al., 2017; Zheng et al., 2020) primarily focus on global feature alignment between visual and textual representations. Later approaches (Gao et al., 2022; Cao et al., 2025; Shao et al., 2022; Yang et al., 2024) introduce multi-granularity modeling to facilitate semantic alignment across diverse modalities. Recently, a growing number of works (Bai et al., 2023a; Qin et al., 2024; You et al., 2025; Zhang et al., 2026) leverage general cross-modal knowledge of visual-language pretraining models (Radford et al., 2021; Li et al., 2021; 2022) for this fine-grained retrieval task. Furthermore, recent works (Bai et al., 2023b; Jiang et al., 2025; Zuo et al., 2024a; Tan et al., 2024; Yang et al., 2023) leverage MLLMs (Liu et al., 2024; Fang et al., 2025; Liang et al., 2025; Fang et al., 2026; Rong et al., 2025; Ye et al., 2025) to automatically generate textual descriptions for person images, enabling the construction of large-scale pretraining datasets that lead to significant performance improvements.

Despite these advancements, existing methods primarily focus on cross-modal alignment while overlooking the limitations of incomplete or ambiguous textual queries. In contrast, our approach introduces a multimodal interactive retrieval paradigm, enabling users to iteratively refine queries and correct retrieved candidates.

### 2.2. Interactive Cross-Modal Retrieval

Interactive cross-modal retrieval leverages user-system interaction to refine queries and improve retrieval accuracy across modalities, which has shown significant advancement and applications in the fields of text-to-image retrieval (Levy et al., 2023; Guo et al., 2018; Cai et al., 2021) and text-to-video retrieval (Liang & Albanie, 2023; Madasu et al., 2022). Existing interactive methods can be broadly grouped into two categories: question-answering interaction and visual feedback interaction. Question-answering interaction (Levy et al., 2023; Lee et al., 2024; Nie et al., 2021) focuses on refining queries by prompting users to supplement missing details or clarify ambiguous information. For example, ChatIR (Levy et al., 2023) leverages large language models to generate the next question based on dialogue history. On the other hand, visual feedback interaction (Chen et al., 2025; Wu et al., 2021a; Wei et al., 2023) allows users to actively articulate distinctions between retrieval candidates and target images. For instance, FashionIQ (Wu et al., 2021a) facilitates interactive refinement by encouraging users to describe key differences during each iteration.

In contrast to the aforementioned methods, our proposed paradigm, MInterPR, presents two significant advancements: **(1)** Most existing approaches focus solely on a single interaction mode, while MInterPR supports both question-answering and visual feedback interactions, offering greater versatility in real-world scenarios. **(2)** Prior works primarily focus on general retrieval scenarios, whereas MInterPR is specifically designed for person retrieval, a challenging fine-grained task where nuanced details play a critical role.

## 3. Benchmark

In this section, we delineate the construction of MInter-PEDES, where each person image (collected from Chat-Pedes (Bai et al., 2025)) is paired with two multi-turn multimodal conversations. The construction involves three steps: reference image sampling, visual feedback generation, and multimodal conversation construction.

### 3.1. Reference Image Sampling

As the dialogue progresses and users provide additional feedbacks, the candidates are iteratively refined, allowing the system to progressively narrow down to the target image. To simulate this process, we leverage TBPS-CLIP (Cao et al., 2024) to guide the reference image sampling. Specifically, given a target image $I_t$ with identity label $y_t$, we first calculate the visual similarity with each image $I_i$ in the dataset through their normalized visual representations:

$$s(I_i, I_t) = \frac{E_I(I_i) \cdot E_I(I_t)}{\|E_I(I_i)\|\|E_I(I_t)\|}, \tag{1}$$

where $E_I$ denotes the visual encoder. The similarity scores are then normalized as sampling probabilities through a softmax function:

$$p(I_i|I_t) = \begin{cases} \dfrac{\exp(s(I_i, I_t))}{\sum_{y_j \neq y_t} \exp(s(I_j, I_t))}, & y_i \neq y_t \\ 0, & y_i = y_t \end{cases} \tag{2}$$

where we enforce $p(I_i|I_t) = 0$ when $y_i = y_t$ to exclude images with the same identity. Subsequently, two sampling strategies are applied to select reference images:
(1) **Top-K Sampling**: This strategy selects $K$ reference images with the highest sampling probabilities, ensuring strong visual correlation with the target image:

$$\mathcal{R}_t^{\text{top-k}} = \{I_i \mid i \in \text{top-}K(p(I_i|I_t))\}. \tag{3}$$

(2) **Probabilistic Sampling**: This strategy selects $K$ reference images according to the probabilities $p(I_i|I_t)$, thereby preserving randomness and diversity:

$$\mathcal{R}_t^{\text{prob}} \sim \text{Multinomial}(K; p(I_i|I_t)), \tag{4}$$

where Multinomial$(\cdot)$ denotes multinomial distribution without replacement. As a result, each target image is associated with two sets of $K$ reference images, corresponding to retrieval candidates at each of dialogue rounds. In this paper, we set $K = 2$ for each dialogue, as we found that increasing the number of turns yields only marginal improvements, while significantly increasing resource consumption and training/inference time, as detailed in Appendix C.

*Table 1.* Statistics of our constructed dataset compared to existing TPR and ChatPR benchmarks. (Abbreviations: Anno = Annotation, QA = Question Answering, MM = Multimodal.)

| Dataset | Data Scale | | | | Word Number | | | Supported Task | | |
|---|---|---|---|---|---|---|---|---|---|---|
| | #ID | #Image | #Anno | Anno Type | Max | Min | Avg | TPR | ChatPR | MInterPR |
| CUHK-PEDES (Li et al., 2017) | 13,003 | 40,206 | 80,440 | Text | 96 | 15 | 23.5 | ● | ○ | ○ |
| ICFG-PEDES (Ding et al., 2021) | 4,102 | 54,522 | 54,522 | Text | 83 | 9 | 37.2 | ● | ○ | ○ |
| RSTPReid (Zhu et al., 2021) | 4,101 | 20,505 | 41,010 | Text | 70 | 11 | 26.5 | ● | ○ | ○ |
| UFine6926 (Zuo et al., 2024b) | 6,926 | 26,206 | 52,412 | Text | 218 | 30 | 80.8 | ● | ○ | ○ |
| SYNTH-PEDES (Zuo et al., 2024a) | 312,321 | 4,791,771 | 12,138,157 | Text, Attribute | 62 | 8 | 23.7 | ● | ○ | ○ |
| ChatPedes (Bai et al., 2025) | 12,003 | 37,128 | 74,256 | QA Dialogue | 388 | 12 | 114.4 | ● | ● | ○ |
| Interactive-PEDES (Lu et al., 2025) | 13,051 | 54,749 | 54,749 | QA Dialogue | 193 | 75 | 135.4 | ● | ● | ○ |
| MInterPEDES (Ours) | 12,003 | 37,128 | 74,256 | MM Dialogue | 474 | 47 | 183.3 | ● | ● | ● |

## 3.2. Corrective Feedback Generation

To simulate corrective feedbacks on how the visual appearance of retrieved candidates should be adjusted towards the target individual, this step focuses on generating descriptions of visual differences between the reference and target images. Considering the prohibitive cost of annotating large-scale data via human crowd-sourcing, we leverage advanced MLLMs to automatically generate the difference descriptions, guided by the carefully designed instructions detailed in Appendix D.1. Although GPT-4o (OpenAI, 2024) demonstrates superior performance in generating high-quality visual difference descriptions, the substantial computational cost limits its large-scale generation. Therefore, we first employ GPT-4o to generate visual difference descriptions for a small set of 1,000 reference-target image pairs, which are then used to fine-tune the open-source MLLM, InternVL 2.5-8B (Chen et al., 2024), thus enabling scalable and cost-effective generation across the entire dataset.

## 3.3. Multimodal Conversation Construction

Typically, as the dialogue advances, the system is provided with increasingly informative feedback, leading to progressively accurate retrieval. To construct such dialogues, for each target image, we sort the sampled $K$ reference images in ascending order according to their similarity scores computed in Equation 1. Each sorted reference image, together with its associated corrective feedback, constitutes a turn of multimodal dialogue. To simulate real-world scenarios in which the system actively queries users to supplement missing information, we incorporate the question-answering dialogue data from ChatPedes (Bai et al., 2025) at the beginning of the visual feedback dialogues, thereby forming multi-turn multimodal conversations. Furthermore, we conduct meticulous manual curation on the constructed data to ensure the quality and reliability of our dataset. Specifically, we systematically review the dialogue samples to identify and rectify inappropriate cases, including generation errors and incomplete feedback, as detailed in Appendix D.2. This curation process not only eliminates noise but also ensures that the benchmark faithfully reflects real-world scenarios.

Finally, we construct a multimodal conversational dataset, MInterPEDES. As shown in Table 1, compared with existing TPR and ChatPR datasets, MInterPEDES incorporates initial descriptions, question-answering interactions and multimodal dialogues, thereby supporting a broader range of retrieval tasks. More details are provided in Appendix D.

## 4. Method

### 4.1. Task Definition

Multimodal interactive person retrieval (MInterPR) is a multimodal conversational retrieval task, where a user collaborates with an AI system to iteratively identify the target person image $I_t$ from a large image gallery $\mathcal{G}$. The retrieval process can be structured as a multi-turn dialogue:

$$\mathcal{D} = \{T, (Q_1, A_1), \cdots, (Q_M, A_M), (R_1, F_1), \cdots, (R_N, F_N)\}. \quad (5)$$

The dialogue $\mathcal{D}$ comprises three components:

(1) An initial textual description $T$, provided by the user at the beginning of the dialogue, which typically conveys a coarse-grained description for the target person.

(2) A sequence of question-answering pairs $\{(Q_i, A_i)\}_{i=1}^M$, in which the system actively asks clarifying questions about ambiguous or missing attributes, and the user responds with short factual answers to enhance the completeness or clarify the ambiguity of the query.

(3) A sequence of reference image and corrective feedback pairs $\{(R_i, F_i)\}_{i=1}^N$, where at each dialogue round $i$, the system retrieves a top-ranked candidate $R_i \in \mathcal{G}$ according to dialogue history, and the user provides corrective feedback $F_i$ describing how visual attributes of the current candidate $R_i$ should be modified towards the target person image $I_t$.

Through this interactive process, the system progressively narrows down to the target individual by leveraging the accumulated dialogue context across turns. In particular,

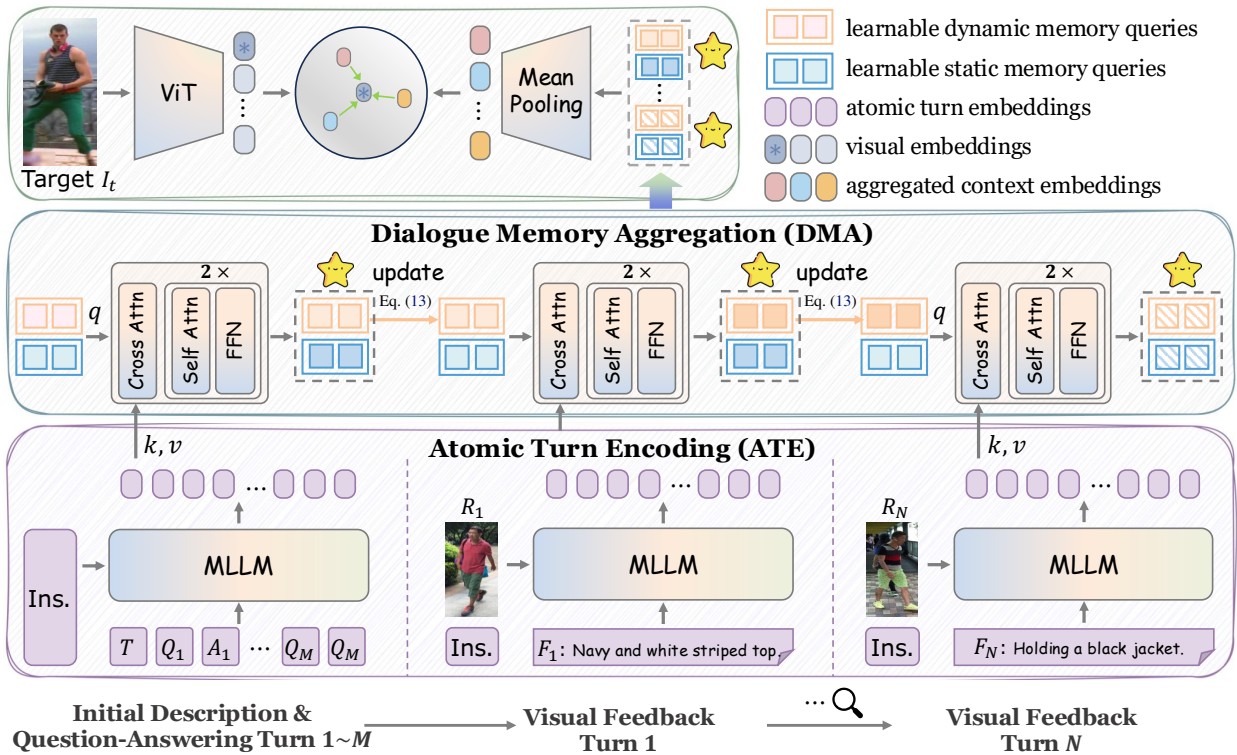

*Figure 2.* The overall framework of MNEMO, which adopts a hierarchical architecture consisting of an Atomic Turn Encoding (ATE) module and a Dialogue Memory Aggregation (DMA) module. ATE individually encodes the multimodal semantic representation of each conversational turn to avoid the computational inefficiency caused by lengthy context. Subsequently, DMA aggregates and contextualizes the turn-level semantics through two groups of learnable memory queries: static queries for capturing shared semantic patterns and fine-grained information, and dynamic queries for tracking and propagating conversational dependencies throughout the dialogue. "Ins." denotes the instruction designed to guide the model to focus on the subtle details of the conversation.

MInterPR exhibits strong generality by encompassing existing person retrieval paradigms: when the number of visual feedback rounds $N = 0$, this task degenerates into ChatPR (Bai et al., 2025); when both $M = 0$ and $N = 0$, MInterPR further reduces to classical TPR (Li et al., 2017).

### 4.2. MNEMO

We propose a multimodal conversational memory-enhanced framework (MNEMO) for MInterPR, which adopts a hierarchical architecture with: (1) an Atomic Turn Encoding (ATE) module to independently encode each dialogue turn, and (2) a Dialogue Memory Aggregation (DMA) module to capture fine-grained information and cross-turn dependencies. The overall framework is illustrated in Figure 2. Detailed notation and algorithm are provided in Appendix A.

#### 4.2.1. PRELIMINARY

MNEMO adopts the open-source MLLM, InternVL 2.5-1B (Chen et al., 2024), as the backbone, which follows a widely used ViT-MLP-LLM architecture comprising: (1) a visual encoder ViT to extract patch-level features from input images, (2) a lightweight MLP-based projection head

to inject image features into the language embedding space, and (3) a pretrained LLM that fuses visual features and contextual language representations for compositional vision-language understanding.

Formally, taking the $i$-th turn of visual feedback $(R_i, F_i)$ as example, the reference image $R_i$ is first fed into ViT to obtain visual embeddings:

$$[\mathbf{v}_i^{\text{cls}}; \mathbf{V}_i] = \text{ViT}(R_i), \quad \mathbf{v}_i^{\text{cls}} \in \mathbb{R}^{d_v}, \ \mathbf{V}_i \in \mathbb{R}^{P \times d_v}, \quad (6)$$

where $\mathbf{v}_i^{\text{cls}}$ denotes the [CLS] token embedding summarizing the entire image representation, $\mathbf{V}_i$ represents the patch embeddings, $P$ is the number of image patches, and $d_v$ is the feature dimension. The patch embeddings are then projected into the language embedding space via an MLP:

$$\mathbf{H}_i = \text{MLP}(\mathbf{V}_i) \in \mathbb{R}^{P \times d_l}, \quad (7)$$

where $d_l$ is the dimension of the LLM's latent space. To integrate visual information into the dialogue context, the projected visual embeddings $\mathbf{H}_i$ are concatenated with the token embeddings of the corrective feedback $F_i$ for compositional vision-language understanding through the LLM:

$$\mathbf{Z}_i = \text{LLM}([\mathbf{Ins}^{\text{emb}}; \mathbf{H}_i; \mathbf{F}_i^{\text{emb}}]) \in \mathbb{R}^{L \times d_l}, \quad (8)$$

where $\mathbf{Ins}^{\text{emb}}$ denotes the token embeddings of the instruction designed to prompt the model to focus on the subtle details, as shown in Appendix D.1, $\mathbf{F}_i^{\text{emb}}$ represents the token embeddings of the feedback $F_i$, $L$ is the sequence length, and $\mathbf{Z}_i$ denotes the output multimodal representation.

### 4.2.2. ATOMIC TURN ENCODING

Existing dialogue modeling methods (Bai et al., 2025; Lu et al., 2025; Levy et al., 2023) typically concatenate all dialogue turns into a single input sequence for LLM processing. However, as dialogue progresses, the input sequence grows substantially, resulting in computational inefficiencies due to the quadratic complexity of attention in Transformer-based LLMs. Moreover, the inclusion of visual feedback further exacerbates this issue, in which reference images are encoded into numerous visual embeddings, significantly increasing sequence length and rendering this approach impractical for real-world applications.

To address this limitation, we introduce an Atomic Turn Encoding (ATE) module, which processes each turn individually to produce atomic turn embeddings. Since the expansion of input sequence length primarily stems from visual feedback dialogues, we retain the conventional concatenation strategy for initial textual inputs and question-answering dialogues, while introducing ATE to process visual feedback dialogues independently. Specifically, we restructure the multi-turn dialogue $\mathcal{D} = \{D_0, (R_1, F_1), \ldots, (R_N, F_N)\}$, where $D_0 = \{T, (Q_1, A_1), \ldots, (Q_M, A_M)\}$ concatenates the initial textual description and the subsequent question-answering pairs. We use the LLM to directly encode $D_0$:

$$\mathbf{Z}_0 = \text{LLM}(\mathbf{D}_0^{\text{emb}}), \qquad (9)$$

where $\mathbf{D}_0^{\text{emb}}$ denotes the token embeddings of $D_0$. For visual feedback dialogues $\{(R_i, F_i)\}_{i=1}^N$, atomic embeddings for each turn are obtained via Equation 6-8. Finally, the overall dialogue representation $\mathcal{D}$ is represented as:

$$\mathbf{Z} = \{\mathbf{Z}_0, \mathbf{Z}_1, \ldots, \mathbf{Z}_N\}, \qquad (10)$$

where $\mathbf{Z}_i$ denotes the atomic turn embeddings at turn $i$.

### 4.2.3. DIALOGUE MEMORY AGGREGATION

While ATE efficiently processes multimodal dialogue turns, it inherently overlooks the mining of fine-grained cues and cross-turn dependencies, thereby limiting the model's ability to accurately comprehend the entire multimodal dialogue. To address this, we propose a Dialogue Memory Aggregation (DMA) module to adaptively capture nuanced details and dynamically integrate contextual information across dialogue turns. Specifically, DMA incorporates two complementary types of memory queries: **(1) Static memory queries**, implemented as learnable embeddings shared across turns, to capture both common semantic patterns and fine-grained information within each turn; **(2) Dynamic memory queries**, designed to dynamically track and propagate evolving dialogue states, for effective modeling of dependencies across sequential dialogue turns.

Formally, DMA integrates static memory queries $\mathbf{Q}^{\text{sta}} \in \mathbb{R}^{N_{\text{sta}} \times d_l}$ shared across dialogue turns, and dynamic memory queries $\mathbf{Q}_i^{\text{dyn}} \in \mathbb{R}^{N_{\text{dyn}} \times d_l}$ updated in each turn $i$. For the initial turn ($i = 0$), the dynamic query $\mathbf{Q}_0^{\text{dyn}}$ is implemented as learnable embeddings. DMA applies a computationally efficient structure with a single multi-head cross attention (Yang et al., 2021) (denoted as $\mathcal{C}$) followed by 2-layer Transformer (Vaswani et al., 2017) blocks (denoted as $\mathcal{T}$) for the interaction between memory queries and atomic turn embeddings:

$$[\mathbf{O}_i^{\text{sta}}; \mathbf{O}_i^{\text{dyn}}] = \mathcal{T}\big(\mathcal{C}([\mathbf{Q}^{\text{sta}}; \mathbf{Q}_i^{\text{dyn}}], \mathbf{Z}_i)\big), \qquad (11)$$

where $\mathbf{O}_i^{\text{sta}} \in \mathbb{R}^{N_{\text{sta}} \times d_l}$ and $\mathbf{O}_i^{\text{dyn}} \in \mathbb{R}^{N_{\text{dyn}} \times d_l}$ denote the output representations of corresponding to the static and dynamic memory queries at turn $i$, respectively. Subsequently, a mean-pooling layer is applied over the static and dynamic memory outputs to obtain the aggregated context embeddings $\mathbf{h}_i$ at turn $i$:

$$\mathbf{h}_i = \frac{1}{N_{\text{sta}} + N_{\text{dyn}}}\Big(\sum_{j=1}^{N_{\text{sta}}} \mathbf{O}_{i,j}^{\text{sta}} + \sum_{k=1}^{N_{\text{dyn}}} \mathbf{O}_{i,k}^{\text{dyn}}\Big). \qquad (12)$$

The dynamic memory queries are then updated using the dynamic output representations to propagate contextual information to the next turn $(i + 1)$:

$$\mathbf{Q}_{i+1}^{\text{dyn}} = \mathbf{O}_i^{\text{dyn}}. \qquad (13)$$

Through this iterative aggregation process, DMA produces a set of contextualized turn-level representations $\mathbf{h} = \{\mathbf{h}_0, \mathbf{h}_1, \ldots, \mathbf{h}_N\}$, which encapsulate both shared semantic structures and evolving conversational context, thereby facilitating holistic dialogue understanding.

### 4.2.4. OPTIMIZATION OBJECTIVES

For the target image $I_t$, the visual encoder ViT of InternVL is employed to obtain the visual embeddings $[\mathbf{v}_t^{\text{cls}}; \mathbf{V}_t] = \text{ViT}(I_t)$. In practical conversational retrieval, an efficient system should direct the user to the target identity within as few dialogue turns as possible. To this end, we explicitly align the aggregated context embeddings of each dialogue turn with the global visual feature of the target image. Formally, the optimization objective is formulated as:

$$\mathcal{L} = \sum_{i=0}^N \mathcal{L}_{\text{nitc}}(\mathbf{h}_i, \mathbf{v}_t^{\text{cls}}) + \mathcal{L}_{\text{sdm}}(\mathbf{h}_i, \mathbf{v}_t^{\text{cls}}), \qquad (14)$$

where $\mathcal{L}_{\text{nitc}}$ represents the normalized image-text contrastive (N-ITC) loss (Cao et al., 2024), and $\mathcal{L}_{\text{sdm}}$ denotes the similarity distribution matching (SDM) loss (Jiang & Ye, 2023).

*Table 2.* Comparison with state-of-the-art methods on MInterPEDES under various retrieval scenarios. $M$ denotes the number of question-answer turns, and $N$ represents the number of visual feedback rounds, as detailed in Task Definition (Section 4.1). "-" indicates the methods not supporting compositional vision-language understanding for MInterPR. Please refer to Section 5.1 for detailed analysis.

| Method | Venue | TPR ($M = 0, N = 0$) | | | | ChatPR ($M \neq 0, N = 0$) | | | | MInterPR ($M \neq 0, N \neq 0$) | | | |
|---|---|---|---|---|---|---|---|---|---|---|---|---|---|
| | | R-1 | R-5 | R-10 | mAP | R-1 | R-5 | R-10 | mAP | R-1 | R-5 | R-10 | mAP |
| IRRA (Jiang & Ye, 2023) | CVPR'23 | 29.15 | 42.24 | 48.33 | 28.24 | 48.26 | 67.96 | 75.65 | 44.64 | - | - | - | - |
| TBPS-CLIP (Cao et al., 2024) | AAAI'24 | 30.32 | 43.59 | 49.25 | 28.54 | 50.21 | 69.27 | 76.43 | 45.44 | - | - | - | - |
| RDE (Qin et al., 2024) | CVPR'24 | 31.05 | 43.88 | 49.56 | 29.85 | 52.77 | 70.61 | 77.47 | 47.98 | - | - | - | - |
| DiaNA (Bai et al., 2025) | CVPR'25 | 29.25 | 43.09 | 49.06 | 28.78 | 75.67 | 90.68 | 94.71 | 66.89 | - | - | - | - |
| SPRC (Bai et al., 2024) | ICLR'24 | 29.42 | 42.95 | 48.69 | 28.43 | 67.63 | 83.47 | 89.62 | 58.95 | 83.13 | 96.23 | 98.54 | 72.81 |
| ICL (Qin et al., 2025) | CVPR'25 | 31.12 | **43.92** | 49.60 | 29.86 | 73.56 | 87.68 | 93.14 | 64.59 | 88.73 | 95.56 | 97.38 | 77.53 |
| LLaVA-ReID (Lu et al., 2025) | ICML'25 | **31.26** | 43.67 | 49.42 | 29.02 | 71.65 | 86.12 | 91.78 | 62.75 | 87.01 | 94.29 | 96.60 | 75.94 |
| **MNEMO (Ours)** | - | 30.89 | **43.92** | **49.64** | **29.89** | **78.09** | **92.27** | **95.48** | **69.18** | **92.26** | **98.86** | **99.63** | **80.31** |

This turn-wise alignment encourages the model to approach the target representation at each dialogue turn, thereby facilitating efficient convergence in multimodal interactive person retrieval scenarios.

## 5. Experiments

In this section, we conduct extensive experiments on MInterPEDES to verify the effectiveness of our method. The implementation details and evaluation protocol are provided in Appendix B. More experiments on TPR benchmarks (Li et al., 2017; Ding et al., 2021; Zhu et al., 2021) and real-user evaluation are presented in Appendix E and F, respectively.

### 5.1. Comparison with State-of-the-Art Methods

We compare MNEMO against a series of state-of-the-art baselines on MInterPEDES under three retrieval scenarios in Table 2: **(1) TPR**: In the classical setting of one-shot textual query, our method achieves competitive performance (*e.g.*, the best mAP of $29.89\%$) despite being designed for multimodal dialogue, verifying its effective textual encoding ability. **(2) ChatPR**: When question-answering dialogues are incorporated, MNEMO significantly outperforms all baselines. For instance, MNEMO surpasses DiaNA (Bai et al., 2025) by $2.42\%$ on R-1 and $2.29\%$ on mAP, demonstrating its strong dialogue modeling compatibility. Unlike existing concatenation approaches, our proposed memory queries act as active dialogue state trackers, effectively integrating interactive information to disambiguate the query. **(3) MInterPR**: In the multimodal conversational interaction setting, MNEMO exhibits a substantial superiority, which we attribute to the fundamental difference in the interaction paradigm. In contrast to existing methods like ICL (Qin et al., 2025) and LLaVA-ReID (Lu et al., 2025) that leverage retrieval candidates as visual prompts for question generation, MNEMO enables users to provide direct corrective feedback on these candidates. As a result, MNEMO demonstrates a more effective convergence to the target identity.

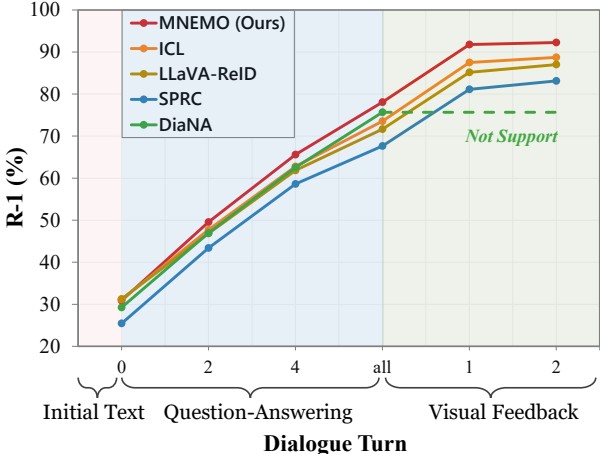

*Figure 3.* Comparison of retrieval performance over dialogue turns. DiaNA and RDE utilize independent encoding for each unimodal input, not supporting compositional vision-language understanding in visual feedback dialogues, while MNEMO demonstrates strong adaptability and achieves superior performance. Please refer to Section 5.2 for detailed analyses.

### 5.2. Performance over Dialogue Turns

We investigate retrieval performance over dialogue turns to evaluate the impact of the incremental conversations under various retrieval scenarios. The comparison illustrated in Figure 3 yields the following observations: **(1) MNEMO achieves superior contextual integration as the dialogue progresses**: While retrieval performance is competitive with only the initial text, MNEMO demonstrates an increasingly pronounced advantage under the subsequent conversational process. This trend reflects its effectiveness to capture and utilize the evolving contextual information throughout the dialogue. **(2) MNEMO demonstrates robust generality across diverse scenarios**: Traditional text-based or chat-based approaches, such as DiaNA (Bai et al., 2025), struggle to handle visual feedback in MInterPR, as they lack joint modeling for compositional vision-language inputs and instead encode each modality independently. In con-

*Table 3.* Ablation of key components in MNEMO. "Train." and "Infer." denote training and inference, respectively. Please refer to Section 5.3.1 for detailed analyses.

| ATE | DMA | R-1 | mAP | Time (ms) ↓ | |
|-----|-----|-----|-----|-------------|-------|
| | | | | Train. | Infer. |
| ✗ | ✗ | 91.51 | 78.26 | 661.78 | 63.76 |
| ✓ | ✗ | 88.90 | 76.99 | 176.25 | 22.77 |
| ✓ | ✓ | 92.26 | 80.31 | 189.24 | 24.07 |

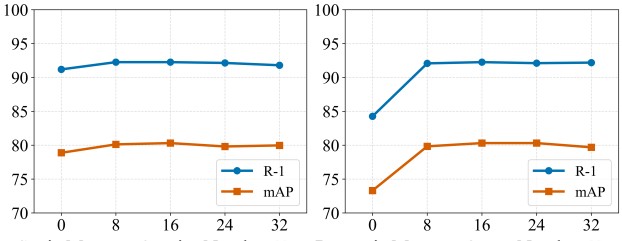

*Figure 4.* Ablation of hyperparameters $N_{sta}$ and $N_{dyn}$. Please refer to Section 5.3.2 for detailed discussion.

trast, MNEMO demonstrates robust performance across all three scenarios, highlighting its adaptability to various person retrieval tasks. **(3) Visual feedback yields significant accuracy improvement**: The introduction of visual feedback significantly boosts retrieval performance for all approaches, underscoring the pivotal role of user-provided corrective feedback within the MInterPR paradigm. These findings demonstrate the necessity of leveraging incremental visual feedback and the integration of evolving context for accurate person retrieval, highlighting its potential for real-world applications. Moreover, additional qualitative visualizations are provided in Appendix G, offering further insights into the effectiveness of MNEMO.

### 5.3. Ablation Study

#### 5.3.1. ABLATION OF KEY COMPONENTS

We conduct an ablation study to evaluate the contributions of ATE and DMA within our MNEMO framework, as shown in Table 3. We summarize key observations as follows: **(1) Concatenation introduces considerable computational overhead**: When both components are removed, the model simply concatenates the entire dialogue into a single long sequence, following prior works (Bai et al., 2025; Lu et al., 2025; Levy et al., 2023). Although this yields relatively strong performance, it introduces significant computational overhead, limiting its application in real-world person retrieval scenarios. **(2) ATE greatly enhances efficiency at the cost of cross-turn dependency modeling**: Instead of directly modeling long-range context for the entire dialogue, ATE processes each dialogue turn separately, which significantly improves computational efficiency during both training and inference. However, this approach fails to capture cross-turn dependencies, resulting in a notable decrease in retrieval accuracy. **(3) DMA boosts performance through contextual aggregation across dialogue turns**: The integration of DMA and ATE not only maintains the efficiency benefits of turn-level encoding but also achieves the best retrieval performance with 92.26% on R-1 and 80.31% on mAP. This demonstrates that DMA is crucial for capturing fine-grained semantic information and aggregating contextual information across dialogue turns.

#### 5.3.2. ABLATION OF HYPERPARAMETERS

We ablate the number of static and dynamic memory queries in DMA in Figure 4: **(1) Static memory queries**: Increasing the number of static memory queries $N_{sta}$ from 0 to 16 consistently improves performance, indicating their enhanced ability to attend to shared inherent structures among dialogue turns. This suggests that static queries play an important role in capturing common patterns within dialogues. However, further increasing $N_{sta}$ beyond 16 brings no additional benefit, indicating that an excessive number of static queries may introduce redundancy and unnecessary complexity. **(2) Dynamic memory queries**: In contrast, removing dynamic memory queries leads to significant performance degradation, underscoring their effectiveness in modeling contextually evolving dialogue states. Performance steadily increases with $N_{dyn}$ up to 16, as more dynamic queries allow the model to better capture diverse and nuanced dialogue dynamics. However, when more dynamic queries are added, performance begins to decline, which can be attributed to overfitting or interference among redundant queries. Based on these observations, we set $N_{sta} = 16$ and $N_{dyn} = 16$ as our default configuration.

## 6. Conclusion

In this work, we introduce multimodal interactive person retrieval (MInterPR), a novel paradigm that enables users to provide visual feedbacks to refine the retrieved candidates during the conversation. To support this task, we expand existing question-answering dialogue data with simulated visual feedbacks and construct MInterPEDES, the first dataset tailored for MInterPR, addressing the scarcity in multimodal conversational data for person retrieval. Moreover, we propose MNEMO, a multimodal conversational memory-enhanced framework, comprising an atomic turn encoding (ATE) module for turn-level semantic modeling and a dialogue memory aggregation (DMA) module for fine-grained information extraction and context-aware integration of dialogue history. Extensive experiments demonstrate the superior retrieval accuracy and computational efficiency of MNEMO, highlighting its promising potential in real-world person retrieval scenarios.

## Acknowledgements

This work was supported by the National Key Research and Development Program of China under Grant (2026YFE0202100), the National Natural Science Foundation of China under Grant (T2541022, 62476188 and 62361166629), and the Major Project of Science and Technology Innovation of Hubei Province under Grant (2025BEA002). The numerical calculations were supported by the supercomputing system at the Supercomputing Center of Wuhan University.

## Impact Statement

This work introduces multimodal interactive person retrieval (MInterPR), a paradigm that combines multimodal dialogue and iterative feedback for accurate person retrieval. Our method (MNEMO), along with the constructed dataset (MInterPEDES), has the potential to positively impact public safety and surveillance by enabling more precise searches for individuals. However, the adoption of such technologies must also account for potential societal challenges. Enhanced surveillance capabilities could inadvertently invade personal privacy, raising ethical concerns regarding misuse by entities with malicious intent. To address these concerns, comprehensive safeguards will be implemented to regulate access to the dataset and the model. Usage restrictions and guidelines are explicitly outlined to ensure these assets are employed solely for research-driven purposes.

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

# A. Notation and Algorithm

*Table 4.* Notation used in this paper, summarizing key variables and operations in our method.

| Symbol | Description | Equation |
|---|---|---|
| $s(I_i, I_t)$ | Cosine similarity between reference image $I_i$ and target image $I_t$ | Eq. 1 |
| $p(I_i\|I_t)$ | Sampling probability conditioned on target image $I_t$ | Eq. 2 |
| $\mathcal{D}$ | The multimodal dialogue sequence | Eq. 5 |
| $M$ | Total number of question-answering dialogue turns | Eq. 5 |
| $N$ | Total number of visual feedback dialogue turns | Eq. 5 |
| $\mathbf{v}_i^{\text{cls}}$ | Global visual representation of reference image $R_i$ | Eq. 6 |
| $\mathbf{V}_i$ | Visual patch embeddings of reference image $R_i$ | Eq. 6 |
| $\mathbf{H}_i$ | Projected patch embeddings in LLM latent space | Eq. 7 |
| $\mathbf{Z}_i$ | Output multimodal representation of the $i$-th dialogue turn | Eq. 8 |
| $\mathbf{Q}^{\text{sta}}$ | Static memory queries shared among all turns | Eq. 11 |
| $\mathbf{Q}_i^{\text{dyn}}$ | Dynamic memory queries updated at turn $i$ | Eq. 11 |
| $\mathbf{h}_i$ | Aggregated context embedding at turn $i$ | Eq. 12 |

---

**Algorithm 1 MNEMO**

---

**Input:** Dialogue $\mathcal{D} = \{T, (Q_1, A_1), \ldots, (Q_M, A_M), (R_1, F_1), \ldots, (R_N, F_N)\}$, target image $I_t$, learnable static memory queries $\mathbf{Q}^{\text{sta}}$, learnable static memory queries $\mathbf{Q}_0^{\text{dyn}}$

**Output:** Loss $\mathcal{L}$ for model optimization

**Atomic Turn Encoding (Section 4.2.2)**

Encode $D_0 = \{T, (Q_1, A_1), \ldots, (Q_M, A_M)\}$ through LLM: $\mathbf{Z}_0 \leftarrow \text{LLM}(\mathbf{D}_0^{\text{emb}})$      // Eq. 9

**for** $i = 1$ **to** $N$ **do**

    Encode reference image $R_i$ via visual encoder: $[\mathbf{v}_i^{\text{cls}}; \mathbf{V}_i] \leftarrow \text{ViT}(R_i)$      // Eq. 6

    Project visual patch embeddings into LLM space: $\mathbf{H}_i \leftarrow \text{MLP}(\mathbf{V}_i)$      // Eq. 7

    Embed corrective feedback $F_i$ through LLM: $\mathbf{Z}_i \leftarrow \text{LLM}([\mathbf{Ins}^{\text{emb}}; \mathbf{H}_i; \mathbf{F}_i^{\text{emb}}])$      // Eq. 8

**end**

**Dialogue Memory Aggregation (Section 4.2.3)**

  **for** $i = 0$ **to** $N$ **do**

    Fuse memory queries with current dialogue turn: $[\mathbf{O}_i^{\text{sta}}; \mathbf{O}_i^{\text{dyn}}] \leftarrow \mathcal{T}(\mathcal{C}([\mathbf{Q}^{\text{sta}}; \mathbf{Q}_i^{\text{dyn}}], \mathbf{Z}_i))$      // Eq. 11

    Aggregate static and dynamic memory outputs: $\mathbf{h}_i \leftarrow \frac{1}{N_{\text{sta}}+N_{\text{dyn}}}\left(\sum_{j=1}^{N_{\text{sta}}} \mathbf{O}_{i,j}^{\text{sta}} + \sum_{k=1}^{N_{\text{dyn}}} \mathbf{O}_{i,k}^{\text{dyn}}\right)$      // Eq. 12

    Update dynamic memory queries for the next turn: $\mathbf{Q}_{i+1}^{\text{dyn}} \leftarrow \mathbf{O}_i^{\text{dyn}}$      // Eq. 13

**end**

**Optimization Objectives (Section 4.2.4)**

Extract visual embeddings of target image: $[\mathbf{v}_t^{\text{cls}}; \mathbf{V}_t] \leftarrow \text{ViT}(I_t)$

Align each turn with target image: $\mathcal{L} \leftarrow \sum_{i=0}^{N} \mathcal{L}_{\text{nitc}}(\mathbf{h}_i, \mathbf{v}_t^{\text{cls}}) + \mathcal{L}_{\text{sdm}}(\mathbf{h}_i, \mathbf{v}_t^{\text{cls}})$      // Eq. 14

**return** $\mathcal{L}$

---

# B. Implementation Details

We employ InternVL 2.5-1B (Chen et al., 2024) as the backbone of MNEMO. The parameters in DMA are randomly initialized. Following prior works (Yang et al., 2023; Li et al., 2024; Bai et al., 2025), we adopt a two-stage training procedure. MNEMO is first pretrained on the synthetic text-image paired dataset, MALS (Yang et al., 2023), where each annotated text serves as the initial description within the conversation. Subsequently, the model is finetuned on MInterPEDES to adapt to multimodal conversational data. During training, we apply data augmentation strategies, including random horizontal flipping, random resizing with cropping, and random erasing. Additionally, we introduce random masking to user-provided responses with a probability of 15% to enhance the model's robustness. The number of static memory queries $N_{\text{sta}}$ and the number of dynamic memory queries $N_{\text{dyn}}$ are both set to 16. Further settings are detailed in Table 5.

*Table 5.* Training configurations of MNEMO.

| Hyperparameter | Pretraining | Finetuning |
|---|---|---|
| epochs | 2 | 5 |
| global batch size | 256 | 64 |
| image size | $280 \times 140$ | $448 \times 224$ |
| LLM sequence length | 100 | 400 |
| learning rate schedule | cosine decay | cosine decay |
| optimizer | AdamW (Loshchilov, 2017) | AdamW (Loshchilov, 2017) |
| learning rate | 5e-5 | 5e-5 |
| warmup steps | 100 | 50 |
| precision | bf16 | bf16 |
| GPUs for training | $4 \times$ RTX 4090 (48GB) | $4 \times$ RTX 4090 (48GB) |

*Table 6.* Retrieval performance and resource consumption over various visual feedback turns. "Mem." denotes the GPU memory usage during training.

| Turn | R-1 | mAP | Mem. (GB) ↓ | Time (ms) ↓ | |
|---|---|---|---|---|---|
| | | | | Training | Inference |
| 0 | 78.09 | 69.18 | 15.33 | 119.96 | 14.25 |
| 1 | 91.36 | 80.04 | 24.83 | 146.26 | 19.61 |
| 2 | 92.26 | 80.31 | 35.06 | 189.24 | 24.07 |
| 3 | 92.48 | 80.46 | 46.47 | 241.31 | 32.15 |
| 4 | 92.34 | 80.53 | 58.36 | 306.21 | 39.86 |

**Evaluation Protocol**   We adopt the widely-used Rank-K (R-K for short, K$=1, 5, 10$) metric in our experiments, which reflects the percentage of successful retrieval that the ground-truth image is searched within the top K ranked results. In addition, we also employ mean average precision (mAP) as an auxiliary metric to evaluate the overall retrieval performance.

# C. Empirical Analysis of Dialogue Turns

We conduct experiments to investigate the impact of increasing the number of visual feedback turns on retrieval performance and resource consumption. As shown in Table 6, the retrieval accuracy rapidly improves with the first two turns, reaching $92.26\%$ on R-1 and $80.31\%$ on mAP after two turns. However, further increasing the number of turns yields only marginal gains in retrieval performance. We conjecture that this limited improvement with more turns may stem from the fact that most informative feedback is already captured within the first two rounds, and subsequent turns tend to provide redundant information. On the other hand, increasing the number of visual feedback turns inevitably extends the dialogue sequence, substantially increasing GPU memory usage as well as training and inference time. Therefore, to achieve a fair balance between efficiency and effectiveness, we adopt two visual feedback turns as our default benchmark setting. Notably, our framework is flexible and can be readily extended to support more visual feedback turns if required by specific applications.

# D. More Details of MInterPEDES

## D.1. Designed Instructions

Considering the substantial cost and time required to collect large-scale user-provided visual feedback from real scenarios, we utilize advanced multimodal large language models (MLLMs) to automatically generate initial feedbacks, which are then manually checked and corrected to produce a scalable and reliable dataset. We first present the

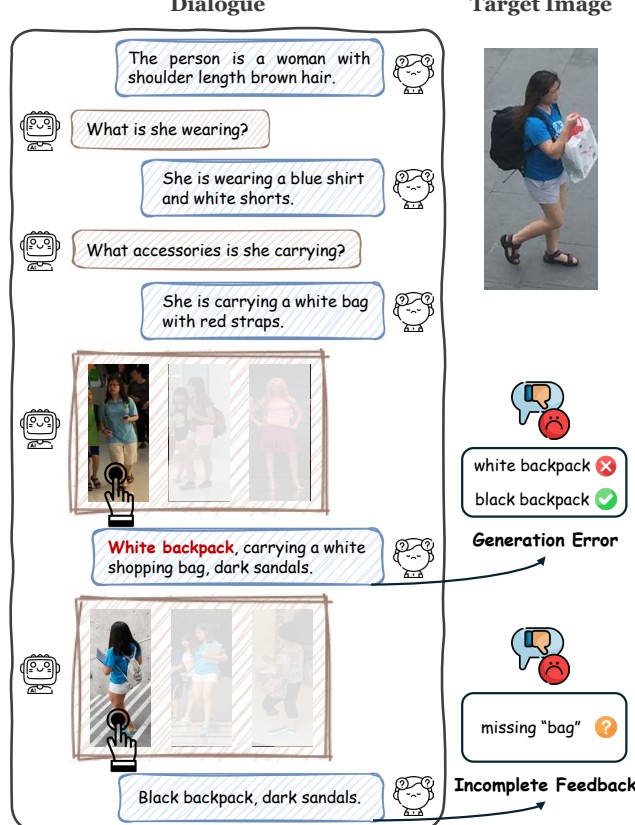

*Figure 5.* Visualization of two typical issues identified during manual curation: generation error and incomplete feedback.

instruction designed for corrective feedback generation for the construction of MInterPEDES in Figure 6. The instruction consists of three parts: task definition, requirements and a set of manually annotated examples, which are designed to fully activate the instruction-following and context-learning capabilities of MLLMs for accurate generation. Additionally, we introduce the system message instruction for MNEMO in Figure 7, which guides the model to focus on detailed nuances of multimodal conversations.

---

**Instruction for Corrective Feedback Generation**

**Task Definition**

- Compare the reference person image and the target person image.
- Identify and describe only the most certain and visible differences in the target image's appearance.

**Requirements**

1. Describe only the target image's appearance based on confirmed differences from the reference image.
2. Focus exclusively on visible appearance changes, including:
    - Head accessories (hat, headband, hairstyle, etc.)
    - Glasses (presence, type, color, etc.)
    - Top clothing (color, type, sleeve length, outerwear, etc.)
    - Bottom clothing (pants, skirt, color, length, type, etc.)
    - Shoes (color, type, style)
    - Accessories (backpack, handbag, scarf, gloves, watch, etc.)
3. Avoid uncertain or ambiguous descriptions—only describe features that are clearly visible and definite.
4. Do not mention the reference image's appearance, only describe the changes in the target image.
5. Only describe differences—if a feature remains the same, do not mention it.
6. Use concise phrases instead of full sentences to describe the target image's appearance.
7. List multiple changes in a single line, separated by commas.

**Example Output**

- No hat, short haircut
- Black round-frame glasses
- Gray hoodie, unzipped, with a white T-shirt inside
- Light gray sweatpants, looser fit
- No backpack, carrying a black crossbody bag
- Brown leather shoes

The reference person image and the target person image are as follows:
- Reference image: <#reference_image>
- Target image: <#target_image>

*Figure 6.* Instruction for corrective feedback generation during the construction of MInterPEDES.

---

**System Message Instruction for MNEMO**

You are an intelligent system designed to retrieve person images from dialogues. Focus on the key attributes such as clothing, accessories, gender, and distinctive features from the conversation.

*Figure 7.* System message instruction for MNEMO, guiding the model to focus on detailed nuances of multimodal conversations.

## D.2. Manual Curation

To ensure the quality and reliability of MInterPEDES, we perform meticulous manual curation on the automatically generated dialogue samples. During this process, we follow strict principles to identify and correct errors through manual annotation. Specifically, we focus on the following issues:

- **Generation Errors:** The generated dialogue is inconsistent with the visual content of the target person image, leading to data noise or even logical conflicts. For instance, as shown in Figure 5, the fourth dialogue turn describes a "white backpack", whereas the target person image actually shows a "black backpack". This mismatch eventually leads to an incorrect retrieval in the next turn.

- **Incomplete Feedback:** The feedback in the dialogue overlooks critical attributes, which may lead to ambiguity or confusion in identifying the target person. For example, in the last turn of Figure 5, the description omits mention of the "bag", causing the system to incorrectly assume that the target image shares the same attributes (*e.g.*, carrying gray books) with the retrieved candidate, potentially resulting in incorrect identification of the target person.

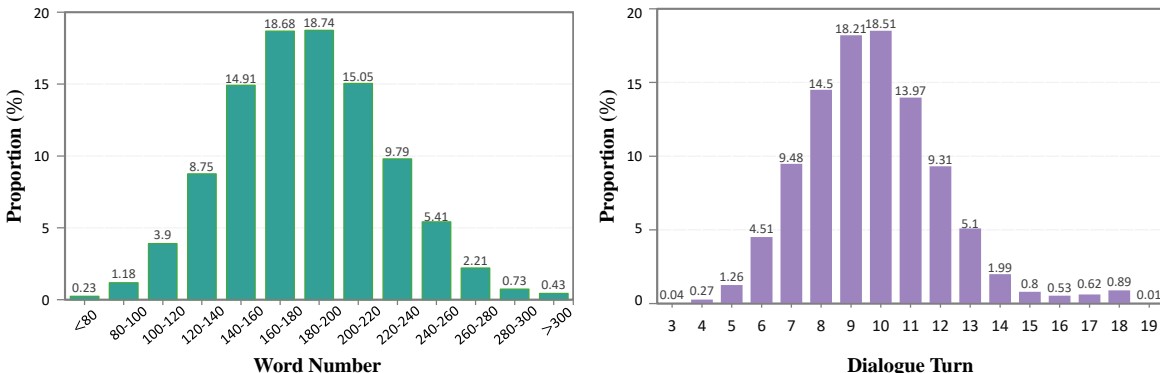

*Figure 8.* Data distribution of word number (*left*) and dialogue turn (*right*) in our dataset.

Through this meticulous curation workflow, we ensure that the final version of MInterPEDES maintains a high standard of logical consistency, semantic correctness and annotation quality. As a result, the dataset offers a trustworthy foundation for MInterPR and facilitates further progress in multimodal interactive person retrieval research.

### D.3. Data Distribution

To provide a comprehensive understanding of our dataset, we analyze the distribution of key attributes, including the number of words and the number of dialogue turns per conversation. As illustrated in Figure 8, the word number per dialogue exhibits a broad range, reflecting the diversity and richness of user expressions captured in our dataset. Similarly, the distribution of dialogue turns demonstrates that our dataset covers both short and long conversations, thereby accommodating various interactive retrieval scenarios. These statistics highlight not only the scale but also the variety present in MInterPEDES, which is essential for training robust and generalizable models for MInterPR. This diverse data distribution is expected to facilitate the development and evaluation of methods that can effectively handle real-world conversational complexity.

### D.4. Data Split

MInterPEDES is a multimodal conversational dataset specifically designed for multimodal interactive person retrieval (MInterPR). The dataset comprises a total of $37,128$ images from $12,003$ identities, sourced from CUHK-PEDE (Li et al., 2017). Each image is paired with two generated dialogues, corresponding to different reference image sampling strategies as detailed in Section 3.1. Following the data split protocol of CUHK-PEDES, we divide $34,054$ images from $11,003$ identities into the training set, and the remaining $3,074$ images from $1,000$ identities into the test set. Overall, MInterPEDES provides a large-scale, diverse, and high-quality resource for advancing research in MInterPR.

## E. Generalization Experiments

MNEMO demonstrates strong adaptability to the TPR task, where each text serves as an initial description within conversations. To evaluate the generalization ability of MNEMO, we conduct extensive experiments on traditional TPR benchmarks, including CUHK-PEDES (Li et al., 2017), ICFG-PEDES (Ding et al., 2021), and RSTPReid (Zhu et al., 2021). Existing TPR approaches can be categorized into two types according to whether performing cross-modal interaction during inference: **Joint Encoder** typically leverages the cross-attention mechanism to perform cross-modal interaction between images and texts; while **Dual Encoder** relies on independent encoding for unimodal data without cross-modal interaction.

We evaluate our method from two aspects. **(1) Retrieval performance:** As shown in Table 7, MNEMO employs a simple Dual Encoder architecture but still achieves promising results. We also observe a gap in performance compared with the best-performing methods, primarily due to the limited scale of our pre-training data. For example, RDE (Qin et al., 2024) employs CLIP, pretrained on 400M text-image paired data, as the backbone, while our MNEMO is only pretrained on MALS (Yang et al., 2023) with 1.5M paired data. **(2) Model parameters and efficiency:** As shown in Table 8, the incorporation of MLLMs introduces additional parameters and a slight increase in inference time, while the substantial performance gains on ChatPR and MInterPR justify this trade-off. Furthermore, MNEMO achieves much higher efficiency than joint-encoder methods by eliminating the need for complex cross-modal fusion modules.

*Table 7.* Comparison with state-of-the-art methods on TPR benchmarks. Existing approaches can be categorized into two types according to whether performing cross-modal interaction during inference: Joint Encoder (w/ cross-modal interaction) and Dual Encoder (w/o cross-modal interaction). Please refer to Appendix E for detailed analyses.

| Method | Venue | CUHK-PEDES | | | | ICFG-PEDES | | | | RSTPReid | | | |
|---|---|---|---|---|---|---|---|---|---|---|---|---|---|
| | | R-1 | R-5 | R-10 | mAP | R-1 | R-5 | R-10 | mAP | R-1 | R-5 | R-10 | mAP |
| *Joint Encoder* | | | | | | | | | | | | | |
| RaSa (Bai et al., 2023a) | IJCAI'23 | 76.51 | 90.29 | 94.25 | 69.38 | 65.28 | 80.40 | 85.12 | 41.29 | 66.90 | 86.50 | 91.35 | 52.31 |
| APTM (Yang et al., 2023) | MM'23 | 76.53 | 90.04 | 94.15 | 66.91 | 68.51 | 82.99 | 87.56 | 41.22 | 67.50 | 85.70 | 91.45 | 52.56 |
| AUL (Li et al., 2024) | AAAI'24 | 77.23 | 90.43 | 94.41 | - | 69.16 | 83.32 | 88.37 | - | 71.65 | 87.55 | 92.05 | - |
| *Dual Encoder* | | | | | | | | | | | | | |
| ViTAA (Wang et al., 2020) | ECCV'20 | 55.97 | 75.84 | 83.52 | - | - | - | - | - | - | - | - | - |
| LapsCore (Wu et al., 2021b) | ICCV'21 | 63.40 | - | 87.8 | - | - | - | - | - | - | - | - | - |
| SRCF (Suo et al., 2022) | ECCV'22 | 64.04 | 82.99 | 88.81 | - | 57.18 | 75.01 | 81.49 | - | - | - | - | - |
| IRRA (Jiang & Ye, 2023) | CVPR'23 | 73.38 | 89.93 | 93.71 | 66.13 | 63.46 | 80.25 | 85.82 | 38.06 | 60.20 | 81.30 | 88.20 | 47.17 |
| BiLMa (Fujii & Tarashima, 2023) | ICCV'23 | 74.03 | 89.59 | 93.62 | 66.57 | 63.83 | 80.15 | 85.74 | 38.26 | 61.20 | 81.50 | 88.80 | 48.51 |
| TBPS-CLIP (Cao et al., 2024) | AAAI'24 | 73.54 | 88.19 | 92.35 | 65.38 | 65.05 | 80.34 | 85.47 | 39.83 | 61.95 | 83.55 | 88.75 | 48.26 |
| RDE (Qin et al., 2024) | CVPR'24 | 75.94 | 90.14 | 94.12 | 67.56 | 67.68 | 82.47 | 87.36 | 40.06 | 65.35 | 83.95 | 89.90 | 50.88 |
| DiaNA (Bai et al., 2025) | CVPR'25 | 73.26 | 88.15 | 93.78 | 65.72 | 63.78 | 80.85 | 85.66 | 38.86 | 61.15 | 83.35 | 89.65 | 47.59 |
| MNEMO (Ours) | - | 73.86 | 89.29 | 93.95 | 65.92 | 64.70 | 81.11 | 85.86 | 39.41 | 63.90 | 85.05 | 90.70 | 51.06 |

*Table 8.* Comparison with state-of-the-art methods on CUHK-PEDES across retrieval performance, model parameters, and average inference time per query.

| Method | Venue | R-1 | mAP | Parameters (M) ↓ | Time (ms) ↓ |
|---|---|---|---|---|---|
| *Joint Encoder* | | | | | |
| RaSa (Bai et al., 2023a) | IJCAI'23 | 76.51 | 69.38 | 210.2 | 97.5 |
| APTM (Yang et al., 2023) | MM'23 | 76.53 | 66.91 | 214.5 | 59.1 |
| *Dual Encoder* | | | | | |
| IRRA (Jiang & Ye, 2023) | CVPR'23 | 73.38 | 66.13 | 194.5 | 2.5 |
| RDE (Qin et al., 2024) | CVPR'24 | 75.94 | 67.56 | 153.0 | 2.9 |
| DiaNA (Bai et al., 2025) | CVPR'25 | 73.26 | 65.72 | 1341.6 | 11.0 |
| MNEMO (Ours) | - | 73.86 | 65.92 | 938.7 | 10.4 |

## F. Real-User Evaluation

To further examine the practical applicability of MNEMO in realistic interactive retrieval scenarios, we conduct a small-scale real-user evaluation. Unlike the main experiments, where corrective feedback is synthetically generated by MLLMs, this evaluation uses feedback directly provided by human users during interaction. Specifically, we invite 10 human evaluators to interact with our system on 100 target identities. At each interaction turn, the evaluator observes the retrieved candidate and provides free-form natural-language feedback describing how the candidate differs from the target person. MNEMO then incorporates the feedback to update the retrieval results in the next turn.

*Table 9.* Comparison between synthetic and human feedback in the real-user evaluation. R-1 is reported.

| Feedback Source | Turn 0 | Turn 1 | Turn 2 |
|---|---|---|---|
| Human | 78.09 | 89.56 | 90.14 |
| Synthetic | 78.09 | 91.77 | 92.26 |

Table 9 reports the comparison between synthetic and human feedback. While human feedback yields slightly lower performance than synthetic feedback, MNEMO still shows consistent improvements over turns, suggesting that MNEMO remains effective under real user interactions and is robust to the variability of human-provided corrective feedback.

## G. Qualitative Results

We provide additional qualitative results in Figure 9 and Figure 10, which further illustrates the effectiveness of our method in progressively refining retrieval results across dialogue turns. These visualizations demonstrate how multimodal feedback enables our model to narrow down the retrieved candidates and better align with user intent over the conversation.

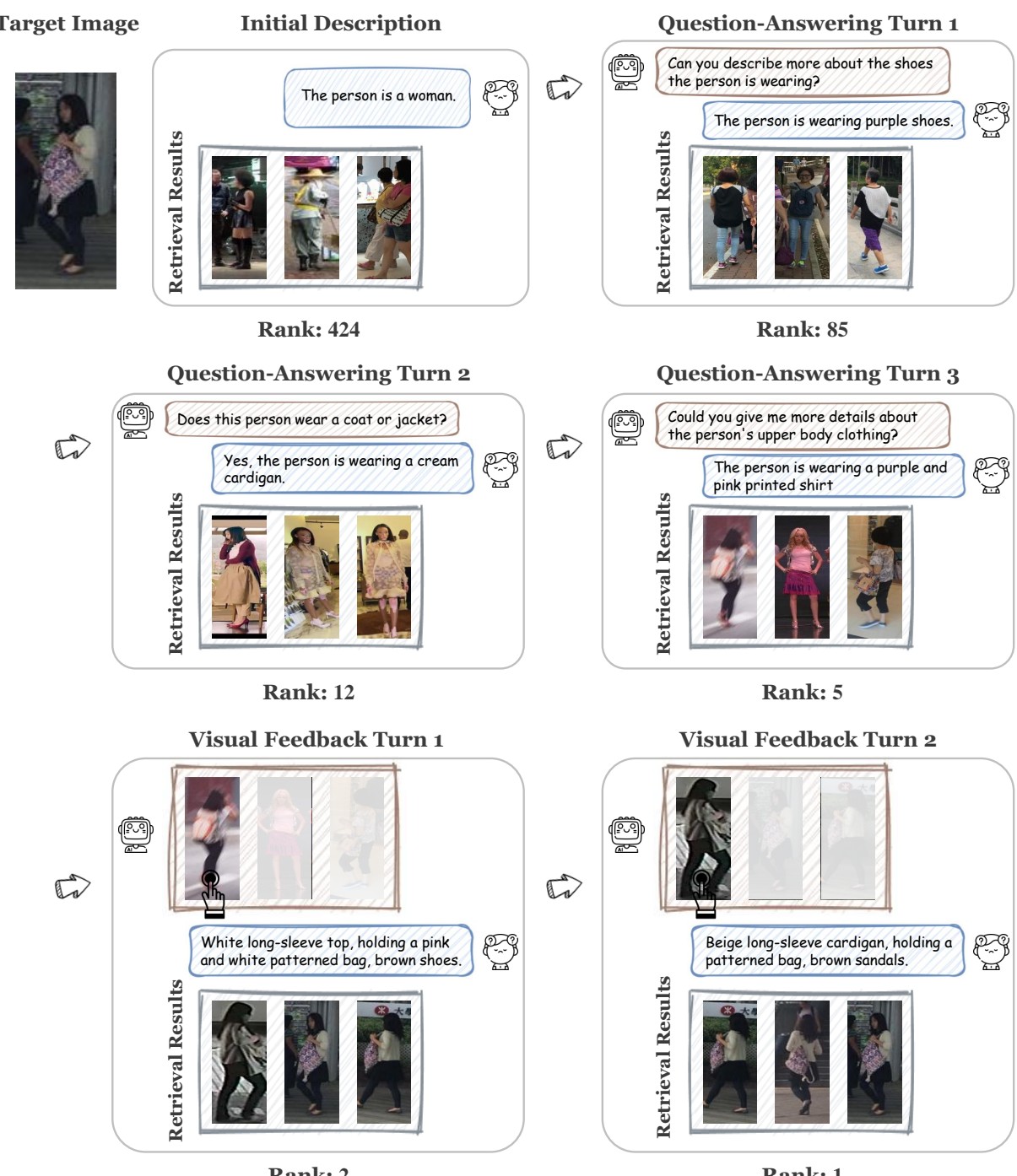

*Figure 9.* Visualization of retrieval results across dialogue turns, showcasing the progressive refinement in retrieved candidates enabled by our method.

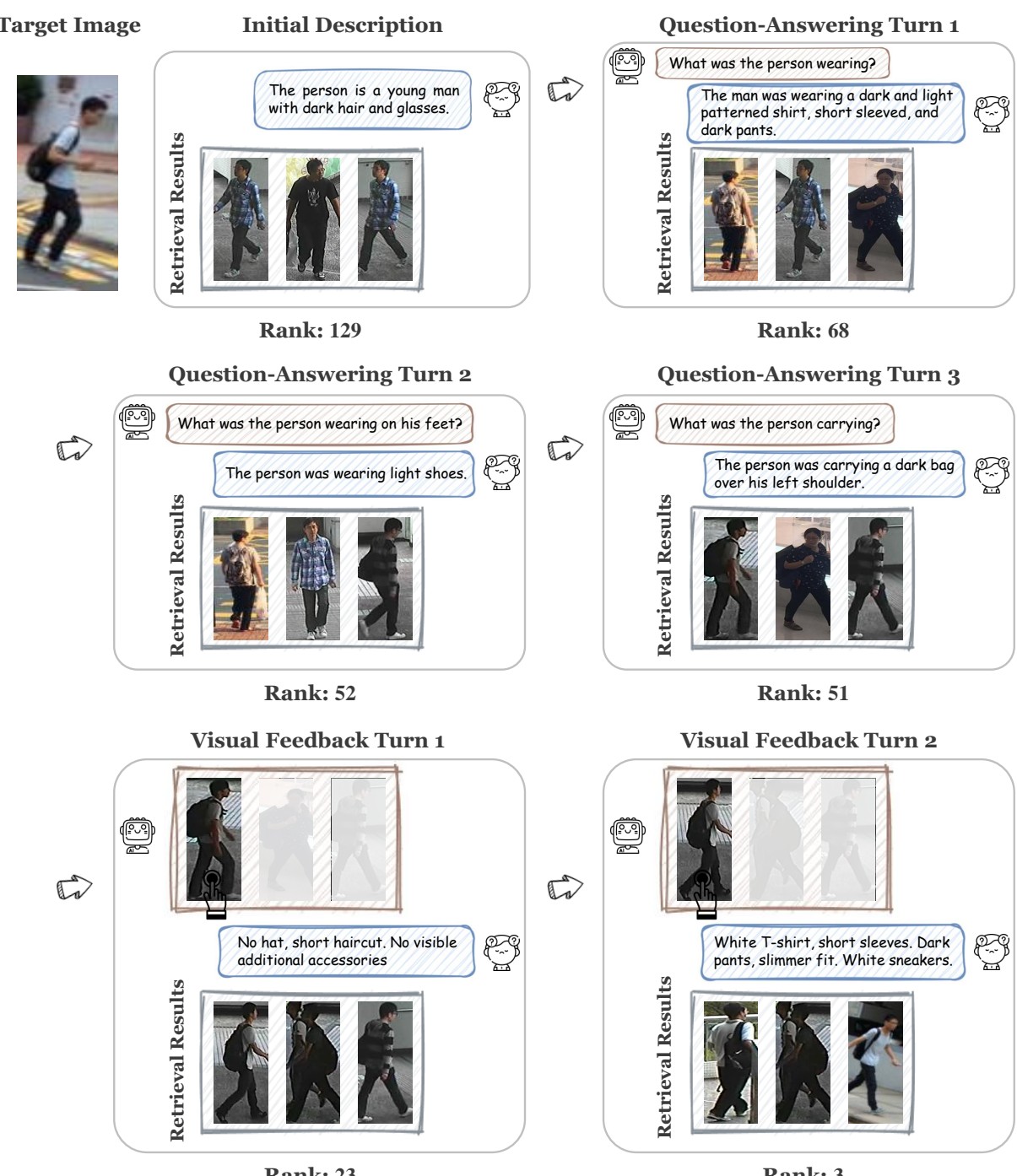

*Figure 10.* Visualization of retrieval results across dialogue turns, showcasing the progressive refinement in retrieved candidates enabled by our method.

