# OpenReview forum: "Interactive Person Retrieval via Multi-Turn Multimodal Conversation"
_ICML.cc/2026/Conference — ICML 2026 regular_

### Official Review · Reviewer_nGFp · 2026-03-07

**Soundness:** 2
**Presentation:** 3
**Significance:** 2
**Originality:** 2
**Overall Recommendation:** 4
**Confidence:** 4

**Summary:**

- The authors propose a new setting for interactive person retrieval in which the system shows a candidate image and the user provides additional corrective feedback based on that image (MInterPR).
- They construct a benchmark (MInterPEDES) and a framework (MNEMO) tailored to this setting. The main difference from existing work is that the system shows the candidate image to the user, i.e., it incorporates visual feedback.
- The MNEMO framework introduces Atomic Turn Encoding (ATE) and Dialogue Memory Aggregation (DMA), where ATE uses retrieved images during the visual-feedback stage.

**Compliance With Llm Reviewing Policy:**

Affirmed.

**Final Justification:**

The rebuttal added additional experiments on real user interaction studies, ablation results, and the reproduction clarification. However, I believe the online user interaction study should be conducted more extensively, covering all methods and different phases of dialogue turns, as offline/static feedback differs substantially from online feedback. That said, I recognize this would be difficult to accomplish within the limited rebuttal period. Considering these points, I raise my score to 4.

**Key Questions For Authors:**

See the weaknesses above.

**Limitations:**

Yes.

**Strengths And Weaknesses:**

Strength

- The proposed method outperforms existing methods in both the ChatPR and MInterPR settings.
- The authors construct a benchmark for the MInterPR setting, including visual feedback for up to two turns.

Weakness

- My biggest concern is that the corrective responses for visual feedback in the benchmark seem to be static, if I understand correctly. In a realistic scenario, when the model shows a candidate image (e.g., a woman with a backpack), the user would provide corrective feedback focused on the differences from that candidate (e.g., “no backpack”). However, the current benchmark appears to construct these corrective feedback responses offline and reuse them, rather than adapting them to the retrieved candidate image.
- I am also unsure where the real performance gains are coming from, since MNEMO already outperforms existing methods in the ChatPR phase. This makes it seem possible that the improvements come primarily from the framework design and/or the higher-quality training dataset, rather than from incorporating visual information during the visual-feedback phase.
In this regard, I have two questions:
    1. Could the authors report results for a variant that optimizes the same loss but does not use retrieved images in ATE during the visual-feedback phase? This would help determine whether the gains in this phase come from the multimodal component or simply from additional textual feedback (especially since the feedback is constructed offline).
    2. Could the authors clarify how the baselines are reproduced? For example, is LLaVA-ReID trained on the same proposed dataset, or are results reported from the original checkpoint? If it is not trained on the same dataset, what are the scores of LLaVA-ReID when fine-tuned on the proposed dataset? If MNEMO does not outperform that setting, then the gains may be due more to the dataset than to the MNEMO design.
- It would also be helpful to report results with the same backbone model fixed across methods (or at least LLaVA-REID). Since MNEMO already outperforms existing methods in the ChatPR phase, I would like to better understand where the improvements actually come from, and whether they truly support the authors’ explanation.

---

> ### Author Rebuttal · Authors · 2026-03-31
>
> We sincerely appreciate that the reviewer recognizes the **strengths of the proposed benchmark** and the **empirical improvements of our method**. Below we respond to each concern in detail.
>
>
>
> ------
>
> **[W1] Offline construction of corrective feedback**
>
> For **fair comparison across methods**, it is necessary to fix the interaction trajectory so that all methods can be evaluated under the same multi-turn dialogue data. Our benchmark is constructed following this principle, which is also a common practice in existing interactive retrieval benchmarks such as FashionIQ [1] and IVCR-200K [2]. Moreover, our benchmark consists of multi-turn interactions with **diverse corrective feedback** arising from different target-candidate discrepancies across turns, as well as **diverse interaction patterns** (including a single text description, question-answering interaction, and visual feedback), which reflects the complexity of online interactive retrieval.
>
> Additionally, we conducted an **online real-user study**, in which 10 human evaluators were asked to identify 100 target persons by providing corrective feedback based on currently retrieved candidates at each turn. As shown in Table J, we observe a consistent trend of performance improvement across turns in both the offline and online settings, indicating that our benchmark provides a meaningful approximation of realistic online interactive retrieval. We will clarify this point more explicitly in the revised manuscript.
>
> **Table J: Comparison with online real-user interaction study (R-1 reported).**
>
> | Setting | Turn 0 | Turn 1 | Turn 2 |
> | ------- | ------ | ------ | ------ |
> | Offline | 78.09  | 91.77  | 92.26  |
> | Online  | 78.09  | 89.56  | 90.14  |
>
> [1] Fashion iq: A new dataset towards retrieving images by natural language feedback. CVPR 2021.
>
> [2] IVCR-200K: A large-scale multi-turn dialogue benchmark for interactive video corpus retrieval. SIGIR 2025.
>
>
>
> ------
>
> **[W2.1] Contribution of retrieved images in the visual-feedback phase**
>
> Thank you for this insightful suggestion. To assess the contribution of the visual information, we evaluated a **text-only** setting that excludes the reference image in ATE during the visual-feedback phase. As shown in Table K, Image+Text consistently outperforms Text-only, indicating that the gains are not solely due to additional textual supervision, while the retrieved reference images provide complementary visual cues that further improve retrieval performance.
>
> **Table K: Effect of using retrieved images during the visual-feedback phase (R-1 reported).**
>
> | Variant             | Turn 0 | Turn 1 | Turn 2 |
> | ------------------- | ------ | ------ | ------ |
> | Text-only           | 78.09  | 89.14  | 90.32  |
> | Image + Text (Ours) | 78.09  | 91.77  | 92.26  |
>
>
>
>
>
> ------
>
> **[W2.2] Clarification on LLaVA-ReID reproduction**
>
> The reported LLaVA-ReID results are **not** from its original checkpoint. Instead, we used the **official codebase** and **re-trained it on our dataset**. To ensure a fair comparison under the **same benchmark protocol**, we make the following necessary data interface adaptations:
>
> (1) we use the reference image as the selected candidate image by the **Selector** in LLaVA-ReID;
>
> (2) we use the corrective feedback as the response that would otherwise be produced by the **Answerer** in LLaVA-ReID;
>
> (3) we re-train the **Retriever** on the concatenated dialogue (initial description + QA pairs) and keep it frozen afterward, consistent with the original settings.
>
> All other settings are kept aligned with the official implementation. We will clarify this reproduction procedure in the revision.
>
>
>
>
>
> ------
>
> **[W3] Same-backbone comparison**
>
> In the main model, we adopt InternVL 2.5 as the backbone because MInterPR requires strong multimodal understanding over multi-turn dialogue. To control for backbone effects, we further conducted a **same-backbone comparison** using CLIP, which is also the Retriever backbone in LLaVA-ReID. Under the CLIP backbone, **ATE** still processes each turn separately. For the visual feedback turn, we concatenate the visual embeddings of the reference image and the textual embeddings of the corrective feedback as the atomic turn embeddings, which are further aggregated by **DMA** in the same way as in our main model. As shown in Table L, MNEMO still outperforms LLaVA-ReID under the same backbone, indicating that the gains are not only due to backbone choice, but also to the proposed turn-wise multimodal encoding and dialogue memory aggregation design.
>
> **Table L: Comparison under the same backbone.**
>
> | Method     | Backbone | R-1   | R-5   | R-10  | mAP   |
> | ---------- | -------- | ----- | ----- | ----- | ----- |
> | LLaVA-ReID | CLIP     | 87.01 | 94.29 | 96.60 | 75.94 |
> | MNEMO      | CLIP     | 89.32 | 96.02 | 97.74 | 78.28 |

---

> > ### Author Rebuttal · Reviewer_nGFp · 2026-04-03
> >
> > Thank you for the additional experiments on real user interaction studies, ablation results, and the reproduction clarification. However, I believe the online user interaction study should be conducted more extensively, covering all methods and different phases of dialogue turns, as offline/static feedback differs substantially from online feedback. That said, I recognize this would be difficult to accomplish within the limited rebuttal period. Considering these points, I raise my score to 4.

---

> > > ### Author Response · Authors · 2026-04-04
> > >
> > > We sincerely appreciate your thoughtful feedback and are glad that the additional experiments and clarifications were helpful.
> > > Your comments have inspired us to recognize the importance of a more extensive online interaction study, and we will explore this in more depth in our future work. Thank you for raising your score to 4, and for your valuable suggestions.

---

### Official Review · Reviewer_jsHy · 2026-03-11

**Soundness:** 3
**Presentation:** 3
**Significance:** 2
**Originality:** 2
**Overall Recommendation:** 4
**Confidence:** 3

**Summary:**

This paper studies a new retrieval setting called multimodal interactive person retrieval (MInterPR). The goal is to identify a target person image through a multi-turn interaction between the user and the retrieval system. Instead of relying only on a single textual query or question–answer interaction, the proposed framework allows users to refine retrieval results by providing visual difference feedback on retrieved candidates.

To support this task, the paper introduces a new dataset MInterPEDES, constructed by extending existing QA-style dialogue datasets with visual feedback turns. For each target image, visually similar reference images are sampled as candidate results, and multimodal large language models are used to generate descriptions of visual differences between the candidate image and the target image. These feedback descriptions are then incorporated into a multi-turn multimodal dialogue. The resulting dataset contains more than 74k multimodal conversations combining initial textual descriptions, QA dialogue, and visual corrective feedback. The paper also proposes MNEMO, a multimodal conversational memory-enhanced retrieval framework. The model contains two main components, Atomic Turn Encoding (ATE) and Dialogue Memory Aggregation (DMA). Experiments on the proposed dataset evaluate three settings: text-only retrieval, chat-based retrieval, and multimodal interactive retrieval. The results show that MNEMO improves retrieval performance across all scenarios, with particularly strong improvements in the multimodal interactive setting.

**Compliance With Llm Reviewing Policy:**

Affirmed.

**Final Justification:**

This paper is a clear with a useful new problem setting, a substantial benchmark, and a reasonable memory-based retrieval architecture, so the soundness, significance, and overall presentation are good. The main concern is that the interaction data is heavily synthesized, the corrective feedback is generated by comparing candidates to the ground-truth target, and the evaluation is still mostly simulated rather than based on real user behavior, even though the task itself is well motivated and the method is carefully designed. The rebuttal helped by adding a small real-user study, a memory-based feedback setting, and clarification on full gallery retrieval and longer dialogues, so it did address part of my concerns. Overall, I view the paper as a meaningful and original contribution, but with limitations in external validity.

**Key Questions For Authors:**

1. Since the corrective feedback descriptions are generated by comparing candidate images with the ground-truth target image, how well do these feedbacks resemble real user feedback in interactive search?

2. The dataset restricts the number of visual feedback rounds to two. Have the authors experimented with longer dialogues, and does the memory aggregation mechanism remain effective in those settings?

3. Recent multimodal LLM agents can perform visual reasoning and dialogue. How would prompting such models directly for interactive retrieval compare with the specialized MNEMO architecture?

**Limitations:**

yes

**Strengths And Weaknesses:**

1. Soundness. The paper proposes a clear formulation of the multimodal interactive person retrieval task. The dialogue structure is well defined and includes three components, including an initial textual description, question–answer clarification turns, and visual feedback turns comparing retrieved candidates with the target image. ATE addresses the computational challenge of modeling long multimodal conversations by encoding each turn separately, while DMA reintroduces cross-turn dependencies through memory queries. This design balances efficiency and contextual modeling, which is a reasonable approach for handling long multimodal dialogues. The training objective combines contrastive alignment and similarity distribution matching losses to align dialogue representations with the target image representation, which fits naturally with the retrieval objective. Experimental evaluation is extensive. The paper compares MNEMO against several recent methods under multiple retrieval settings and provides additional analysis such as performance over dialogue turns and ablation studies on the proposed modules.

2. Presentation. The paper is generally well organized and easy to follow. The motivation for introducing interactive retrieval is clearly illustrated in Figure 1, which contrasts single-query retrieval, chat-based retrieval, and the proposed multimodal interactive retrieval paradigm. The architecture diagram also helps clarify how the turn-level encoding and memory aggregation components interact within the model.

3. Significance. Interactive retrieval is an important direction for practical search systems. In many real-world situations, users cannot provide a complete description of the target individual at the start of the search. Allowing users to iteratively refine the query through dialogue and visual feedback could make retrieval systems more usable. The introduction of a dataset for multimodal conversational person retrieval may also stimulate further research in this area.

4. Originality. The main novelty lies in the combination of multimodal interaction and conversational retrieval for person search. While interactive retrieval and multimodal dialogue modeling have been studied separately, the proposed task integrates question–answer interaction and visual corrective feedback within a unified framework. Some of the architectural components, e.g., hierarchical encoding, memory queries, multimodal transformers, are extensions of existing techniques.

Weaknesses:

1. The corrective feedback in MInterPEDES is generated automatically by multimodal LLMs rather than written by human annotators. Although the authors mention manual filtering, the overall annotation pipeline is largely automated. Automatically generated difference descriptions may follow patterns that differ from how real users describe visual differences during interactive search. A more detailed analysis of annotation quality or human evaluation of the generated feedback would strengthen the dataset contribution.

2. The visual feedback descriptions are generated by comparing reference images with the ground-truth target image. This means that the generated feedback explicitly reflects the difference between the candidate and the correct target. In real interactive retrieval scenarios, users may not have access to the target image while interacting with the system. Instead, they typically rely on memory or partial descriptions of the target. As a result, the dataset may not fully reflect the uncertainty present in real user interactions. Clarifying how this design choice affects the realism of the task would be helpful.

3. The experiments simulate user interactions using pre-generated dialogues rather than evaluating the system with actual users. As a result, the reported improvements mainly reflect performance on a constructed benchmark rather than real interactive behavior. User studies or online interaction experiments would provide stronger evidence that the proposed interaction paradigm improves usability in practice.

4. Each dialogue turn in the dataset involves a small number of candidate images. As a result, the retrieval process may be easier than real-world retrieval scenarios where the search space is much larger. Because the candidate set is constrained during dataset construction, improvements in retrieval accuracy may partly reflect properties of the benchmark rather than the inherent effectiveness of the interaction paradigm. In addition, the dataset limits the number of visual feedback rounds to two, mainly for computational reasons. While this simplifies training and evaluation, it also means that the benchmark does not fully capture longer interactive search sessions.

---

> ### Author Rebuttal · Authors · 2026-03-31
>
> We sincerely thank the reviewer for the thoughtful and constructive comments, and for recognizing the **soundness**, **presentation**, **significance**, and **originality** of our work. Below we respond to the main concerns and questions in detail.
>
>
>
> ---
>
> **[W1] Automatically generated feedback may differ from real user feedback**
>
> To assess feedback realism, we conducted a **real-user study**, where 10 human evaluators were invited to interact with our framework and identify 100 target person images. As shown in Table G, although human feedback is naturally noisier and more diverse, MNEMO still achieves consistent improvements across turns, indicating that the generated feedback provides a practical approximation of real user feedback.
>
> **Table G: Comparison of synthetic and human feedback (R-1 reported).**
>
> | Feedback Source | Turn 0 | Turn 1 | Turn 2 |
> | --------------- | ------ | ------ | ------ |
> | Synthetic       | 78.09  | 91.77  | 92.26  |
> | Human           | 78.09  | 89.56  | 90.14  |
>
>
>
> ---
>
> **[W2] Feedback is generated based on ground-truth target images rather than user memory**
>
> We note that the target–candidate comparison is a **standard and widely-used protocol** in existing retrieval benchmarks, e.g., CUHK-PEDES and FashionIQ. We further conducted a **memory-based evaluation** by perturbing feedback with GPT-4o via attribute removal (simulating **incomplete memory**) and modification (simulating  **memory confusion**). As shown in Table H, although performance decreases under memory-based feedback, our method still achieves consistent improvements across turns, indicating that the method remains effective under imperfect feedback.
>
> **Table H: Comparison with memory-based feedback (R-1 reported).**
>
> | Feedback Source          | Turn 0 | Turn 1 | Turn 2 |
> | ------------------------ | ------ | ------ | ------ |
> | Ground-truth image-based | 78.09  | 91.77  | 92.26  |
> | Memory-based             | 78.09  | 85.72  | 87.65  |
>
>
>
> ---
>
> **[W3] Evaluation with simulated interactions rather than real users**
>
> Thank you for this insightful suggestion. As described in **[W1]**, we conducted a **real-user study** in which human users interacted with the system by providing their own feedback. MNEMO still achieves consistent improvements across turns, suggesting that the gains are not merely an artifact of simulated dialogues.
>
>
>
> ---
>
> **[W4] Candidate set size and visual feedback turns**
>
> **(1) Candidate set size.** We would like to clarify that **our retrieval is conducted over the full test gallery**, consistent with the existing text-based person retrieval benchmark, CUHK-PEDES. Therefore, the task is not simplified to a small candidate set during evaluation.
>
> **(2) Visual feedback turns.** We have experimented with longer dialogues in **Appendix C (Table 6)**, where additional turns bring marginal gains but noticeably higher cost. We therefore adopt two visual feedback turns as a practical trade-off.
>
>
>
> ---
>
> **[Q1] How well do these feedbacks resemble real user feedback in interactive search?**
>
> As shown in **[W1]**, our real-user study indicates that the automatically generated feedback provides a practical approximation of human feedback. In addition, **[W2]** shows that the method remains effective under memory-based feedback, demonstrating its robustness beyond the ground-truth image-based setting.
>
>
>
> ---
>
> **[Q2] Have the authors experimented with longer dialogues, and does the memory aggregation mechanism remain effective in those settings?**
>
> Yes. As discussed in **[W4]**, we have experimented with longer dialogues in **Appendix C (Table 6)**. We further provide an ablation study in Table I, showing that **ATE+DMA** consistently outperforms **ATE** under longer dialogues, indicating that the memory aggregation mechanism remains effective.
>
> **Table I: Effectiveness of DMA under different numbers of visual feedback rounds (R-1 Reported).**
>
> | # Visual Feedback Rounds | ATE   | ATE+DMA |
> | ------------------------ | ----- | ------- |
> | 2                        | 88.90 | 92.26   |
> | 3                        | 89.31 | 92.48   |
> | 4                        | 89.14 | 92.34   |
>
>
>
> ---
>
> **[Q3] How would prompting LLM agents directly for interactive retrieval compared with the specialized MNEMO architecture?**
>
> Recent LLM agents are strong at visual reasoning and dialogue understanding, but they are more naturally used as **rerankers over a small candidate set**, e.g., RankRAG [1] and LamRA [2]. In contrast, MNEMO is designed as a specialized end-to-end retrieval architecture for large-gallery interactive retrieval. Therefore, we view prompted MLLM agents as a complementary reranking baseline, but not a direct replacement for our framework.
>
>
>
> [1] Rankrag: Unifying context ranking with retrieval-augmented generation in llms. NeurIPS 2024.
>
> [2] Lamra: Large multimodal model as your advanced retrieval assistant. CVPR 2025.

---

> > ### Author Rebuttal · Reviewer_jsHy · 2026-04-02
> >
> > Thank you to the authors for the helpful rebuttal. The additional evidence provided in the rebuttal, including the real user study, the memory-based feedback evaluation, and the clarification regarding full gallery retrieval and longer dialogue settings, strengthens the empirical support for the paper. But the memory-based feedback setting is still simulated rather than naturally collected from users, and the discussion of directly prompted multimodal LLM agents is conceptual without a direct empirical comparison. I maintain my original rating.

---

> > > ### Author Response · Authors · 2026-04-04
> > >
> > > Thank you for the thoughtful follow-up. We are encouraged that our additional experiments and clarifications are helpful. Following your insightful suggestion, we further provide new experiments to address the remaining concerns below.
> > >
> > >
> > >
> > > ---
> > >
> > > **1. Human-collected memory-based feedback**
> > >
> > > To better evaluate the more realistic scenario, we additionally conduct a human memory-based study with 10 human evaluators on 100 target person images. Unlike the previous human ground-truth-based study in Table G, the evaluators are asked to observe the target image only **before the interaction starts**, and then provide corrective feedback purely from memory **without further access to the target image**.
> > >
> > > As shown in Table M, compared with the ground-truth image-based setting, the memory-based setting leads to a performance drop. This is consistent with our observations during the study that human memory-based feedback is noisier, often missing fine-grained details and providing incorrect feedback. Nevertheless, **our method still achieves consistent improvements over turns**. This further demonstrates that our framework remains effective under imperfect memory-based user feedback.
> > >
> > > We would like to clarify that this work is an initial step toward multimodal interactive person retrieval under a controlled benchmark. Since **memory-based feedback introduces additional human bias and noise**, which may confound fair evaluation across methods, we view it as a complementary realism-oriented evaluation rather than the benchmark setting.
> > >
> > > **Table M: Comparison of ground-truth-based and memory-based feedback (R-1 reported).**
> > >
> > > | Feedback Source                    | Turn 0 | Turn 1 | Turn 2 |
> > > | ---------------------------------- | ------ | ------ | ------ |
> > > | Synthetic ground-truth image-based | 78.09  | 91.77  | 92.26  |
> > > | Synthetic memory-based             | 78.09  | 85.72  | 87.65  |
> > > | Human ground-truth image-based     | 78.09  | 89.56  | 90.14  |
> > > | Human memory-based                 | 78.09  | 83.04  | 86.33  |
> > >
> > >
> > >
> > > ---
> > >
> > > **2. Prompted multimodal LLM agents**
> > >
> > > Thank you for your insightful feedback. To conduct an empirical comparison, we use Qwen3-VL-2B as the prompted multimodal LLM and perform the reranking in a **list-wise manner**. Specifically, we first use MNEMO to retrieve the **Top-K** candidates from the full gallery, and then prompt Qwen3-VL-2B with the current dialogue query and the candidate images to rerank these Top-K results.
> > >
> > > As shown in Table N, while prompted MLLM reranking can bring marginal gains on a small candidate set, it incurs substantially higher latency. We also observe that increasing Top-K does not consistently improve performance, since a larger candidate pool introduces more distractors while substantially increasing reranking cost. Overall, these results suggest that current prompted multimodal LLM agents are more suitable as complementary rerankers than as standalone solutions for large-gallery interactive retrieval in the proposed MInterPR task.
> > >
> > >
> > >
> > > **Table N: Comparison with prompted multimodal LLM reranking baseline (*Time denotes the average inference latency per query*).**
> > >
> > > | Method              | K | R-1   | mAP   | Time (ms) |
> > > | ------------------- | ----- | ----- | ----- | --------- |
> > > | MNEMO               | -     | 92.26 | 80.31 | 24.07     |
> > > | MNEMO + Qwen3-VL-2B | 8     | 92.56 | 80.46 | 1683.14   |
> > > | MNEMO + Qwen3-VL-2B | 16    | 92.38 | 80.41 | 4246.80   |
> > > | MNEMO + Qwen3-VL-2B | 32    | 91.76 | 79.72 | 9692.55   |
> > >
> > >
> > >
> > > ---
> > >
> > > We hope these additional experiments and clarifications help address the two remaining concerns, and we would greatly appreciate your consideration in your final assessment. We sincerely thank the reviewer again for the constructive feedback and consideration.

---

### Official Review · Reviewer_D99R · 2026-03-13

**Soundness:** 3
**Presentation:** 3
**Significance:** 3
**Originality:** 3
**Overall Recommendation:** 5
**Confidence:** 5

**Summary:**

This paper studies person retrieval under the multi-turn multimodal conversational setting. It introduces a new task, MInterPR, where a user iteratively interacts with the system through multi-turn dialogue to refine the search for a target person image. The interaction includes both question–answer conversations and visual feedback that describes differences between retrieved candidates and the target. To support this task, the authors construct a multimodal conversational dataset, MInterPEDES. The proposed method, MNEMO, encodes each dialogue turn separately and aggregates the information with a memory-based mechanism to capture cross-turn dependencies for retrieval. Extensive experiments on diverse conversational settings demonstrate the effectiveness of the method.

**Compliance With Llm Reviewing Policy:**

Affirmed.

**Key Questions For Authors:**

Please refer to the ***Weaknesses*** section, particularly regarding the dataset construction strategy, dataset comparison and notation clarity.

**Limitations:**

yes

**Strengths And Weaknesses:**

***Strengths***

1. **Presentation.** The paper is clearly written and well organized. The motivation, dataset construction, and method are presented in a logical order, making the overall framework easy to follow.
2. **Significance.** The work explores multimodal interactive person retrieval and introduces a new multi-turn multimodal conversational setting for this task. It also constructs the first multimodal conversational person retrieval dataset, which could facilitate future research on interactive and multimodal retrieval in this domain.
3. **Soundness.** The proposed method adopts a turn-wise encoding strategy combined with an information aggregation mechanism. This design attempts to balance modeling performance and computational efficiency, and the technical formulation is generally reasonable.
4. **Experiments.** The paper reports comprehensive results under multiple settings, including traditional text-based person retrieval (TPR), chat-based person retrieval (ChatPR), and the proposed multimodal interactive person retrieval (MInterPR). The adequate implementation details provided in the appendix also improve the reproducibility of the work.



***Weaknesses***

1. **Dataset construction validation.** During the dataset construction, the paper introduces two reference image sampling strategies (Top-K sampling and probabilistic sampling) to select candidate images for generating visual feedback. While the design appears intuitively reasonable and is consistent with the overall framework, the paper does not provide empirical analysis to justify this design choice. In particular, it would be helpful to understand how different sampling strategies influence the generated feedback quality and the downstream retrieval performance.

2. **Incomplete dataset comparison.** The dataset statistics mainly compare with classical TPR datasets and several ChatPR datasets. However, some recent representative datasets are missing from the comparison, such as the synthetic datasets MALS and SYNTH-PEDES, as well as the ultra-fine granularity dataset UFine6926. Including these datasets would provide a more comprehensive view of how the proposed dataset differs from existing benchmarks.
3. **Notation clarity.** In line 282, the symbols $N_{sta}$ and $N_{dyn}$ appear for the first time without explicit explanation. Readers may need to infer their meanings from the later context, which slightly interrupts the reading flow when first encountering the formulation. Providing clear definitions of these symbols when they are first introduced would improve the readability and clarity of the method description.

---

> ### Author Rebuttal · Authors · 2026-03-31
>
> We sincerely thank the reviewer for the positive feedback of our work, especially for recognizing the **clarity of presentation**, the **significance of the proposed MInterPR task and MInterPEDES dataset**, the **soundness of the method design**, and the **comprehensiveness of the experiments**. Below we respond to the three concerns in detail.
>
>
>
> ---
>
> **[W1] Dataset construction validation**
>
> Thank you for this insightful suggestion. To validate the impact of the two reference image sampling strategies, we conducted an additional ablation study on different reference image sampling strategies, including **Top-K sampling only**, **probabilistic sampling only**, and **their combination (used in our paper)**.
>
> As shown in Table E,  the combined Top-K + probabilistic sampling strategy consistently performs best. This suggests that the hybrid design provides a better balance between **relevance** and **diversity** of reference candidates: Top-K sampling alone tends to select overly similar candidates, which may limit the informativeness of the generated feedback, while probabilistic sampling alone introduces greater diversity but also more noise. Their combination yields more informative corrective feedback and leads to better downstream retrieval performance.
>
> We will include this validation in the revised manuscript to better justify the design of the reference image sampling strategy.
>
> **Table E: Comparison of different reference image sampling strategies.**
>
> | Sampling Strategy     | R-1   | R-5   | R-10  | mAP   |
> | --------------------- | ----- | ----- | ----- | ----- |
> | Top-K                 | 90.75 | 98.57 | 99.51 | 78.24 |
> | Probabilistic         | 90.20 | 97.88 | 99.14 | 78.15 |
> | Top-K + Probabilistic | 92.26 | 98.86 | 99.63 | 80.31 |
>
>
>
> ------
>
> **[W2] Incomplete dataset comparison**
>
> Thank you for pointing this out. We will update Table 1 in the revised manuscript by including additional recent representative datasets (**MALS**, **SYNTH-PEDES**, and **UFine6926**), as shown in Table F, and clarify the differences between them and MInterPEDES.
>
> In particular, while MALS and SYNTH-PEDES are synthetic datasets and UFine6926 focuses on ultra-fine-grained text-based person retrieval, our proposed MInterPEDES is designed for multi-turn multimodal interactive person retrieval, where retrieval is progressively refined through both dialogue-based interaction and visual corrective feedback. To the best of our knowledge, MInterPEDES is the first benchmark tailored to this setting.
>
> We appreciate this suggestion and will revise the comparison accordingly to better clarify the relationship between our dataset and recent benchmarks.
>
> **Table F: Comparison of different benchmarks.**
>
> | Dataset            | #ID       | #Image    | #Anno      | Anno Type   | Max Word Number | Min Word Number | Avg Word Number |
> | ------------------ | --------- | --------- | ---------- | ----------- | --------------- | --------------- | --------------- |
> | CUHK-PEDES         | 13,003    | 40,206    | 80,440     | Text        | 96              | 15              | 23.5            |
> | ICFG-PEDES         | 4,102     | 54,522    | 54,522     | Text        | 83              | 9               | 37.2            |
> | RSTPReid           | 4,101     | 20,505    | 41,010     | Text        | 70              | 11              | 26.5            |
> | **MALS**           | 1,510,330 | 1,510,330 | 1,510,330  | Text        | 76              | 12              | 27.0            |
> | **SYNTH-PEDES**    | 312,321   | 4,791,771 | 12,138,157 | Text        | 62              | 8               | 23.7            |
> | **UFine6926**      | 6,926     | 26,206    | 52,412     | Text        | 217             | 30              | 80.8            |
> | ChatPedes          | 12,003    | 37,128    | 74,256     | QA Dialogue | 388             | 12              | 114.4           |
> | Interactive-PEDES  | 13,051    | 54,749    | 54,749     | QA Dialogue | 193             | 75              | 135.4           |
> | MInterPEDES (Ours) | 12,003    | 37,128    | 74,256     | MM Dialogue | 474             | 47              | 183.3           |
>
>
>
> ------
>
> **[W3] Notation clarity**
>
> Thank you for pointing this out. In the revised manuscript, we will explicitly define these symbols at their first occurrence to avoid interrupting the reading flow, and carefully check the method section for similar notation issues to further improve readability.

---

> > ### Author Rebuttal · Reviewer_D99R · 2026-04-03
> >
> > Thank you for the detailed rebuttal. My main concerns have been fully addressed. In particular, the additional ablation on reference image sampling strategies and the expanded comparison with recent datasets provide stronger empirical support for the proposed benchmark.
> >
> > Overall, the rebuttal further strengthens my confidence that the proposed benchmark and method are well motivated, technically sound, and supported by sufficiently comprehensive experiments. I appreciate the authors' careful revisions and would encourage that these additions be incorporated into the final paper.

---

> > > ### Author Response · Authors · 2026-04-04
> > >
> > > We sincerely appreciate your positive feedback and are grateful for your thorough review. We are glad to hear that your concerns have been fully addressed. We will incorporate these additions into the final paper. Thank you again for your valuable feedback and continued support.

---

### Official Review · Reviewer_SDm5 · 2026-03-13

**Soundness:** 3
**Presentation:** 3
**Significance:** 3
**Originality:** 4
**Overall Recommendation:** 4
**Confidence:** 4

**Summary:**

This paper introduces Multimodal Interactive Person Retrieval (MInterPR), a new retrieval paradigm that extends traditional text-based person retrieval (TPR) and chat-based person retrieval (ChatPR) by allowing users to provide corrective feedback on retrieved candidates through multi-turn multimodal conversations. The key idea is that users can inspect visually similar candidates and describe how they differ from the target person, progressively steering the system toward the correct identity.
To support this task, the authors construct MInterPEDES, a multimodal conversational dataset built by augmenting existing QA dialogues from ChatPedes with synthesized visual feedback. Reference images are sampled based on visual similarity, and MLLMs (GPT-4o for a small seed set, then a fine-tuned InternVL 2.5-8B for the rest) generate difference descriptions between reference and target images. Manual curation is performed to fix errors. The authors also propose MNEMO, a framework optimized by aligning aggregated context embeddings at every turn with the target image's global visual feature using contrastive and distribution matching losses.

**Compliance With Llm Reviewing Policy:**

Affirmed.

**Final Justification:**

After reviewing the rebuttal, I am pleased that the authors have adequately addressed my concerns. The clarifications provided were clear and directly engaged with the issues I raised. I am therefore maintaining my original positive rating.

**Key Questions For Authors:**

mentioned above

**Limitations:**

Yes (Mentioned in Impact Statement section)

**Strengths And Weaknesses:**

**[Strengths]**

**S1. Strong and well-motivated contribution.** The proposed MInterPR paradigm is a natural and practical extension of existing retrieval settings. The formulation elegantly subsumes both TPR (M=0, N=0) and ChatPR (N=0) as special cases

**S2. Clear presentation and useful qualitative results.** Figures 9 and 10 convincingly show retrieval rank improving from hundreds to 1 across dialogue turns, providing intuitive evidence for the approach's effectiveness.

**[Weakness]**

**W1. Synthetic data pipeline lacks rigorous validation.** The core dataset construction relies on GPT-4o generating feedback for only 1,000 pairs, which is then used to fine-tune InternVL 2.5-8B for the remaining data. There is no quantitative analysis of quality degradation through this distillation process — for example, comparing retrieval performance when using GPT-4o-generated feedback throughout versus the distilled model's output. The manual curation step is mentioned but no statistics are provided

**W2.** **No real user evaluation.** All corrective feedback is synthetically generated by MLLMs. Real users would produce feedback that is noisier, more varied in style, and potentially incomplete in different ways than MLLM-generated text. The absence of any user study significantly weakens the paper's claims about practical applicability. Even a small-scale study with actual users providing feedback would strengthen the work considerably.

**W3. Limited backbone scaling analysis.**  Only InternVL 2.5-1B is used. It remains unclear whether ATE and DMA provide consistent benefits with larger MLLMs

---

> ### Author Rebuttal · Authors · 2026-03-31
>
> We sincerely thank the reviewer for the constructive feedback and for recognizing our work as a **strong and well-motivated** contribution (S1) with **clear presentation and useful qualitative results** (S2). We provide the point-to-point response as follows.
>
>
>
> ---
>
> **[W1] Synthetic data pipeline lacks rigorous validation**
>
> **(1) Generator Comparison:** To assess the quality gap between GPT-4o and the distilled InternVL 2.5-8B, we randomly sampled **4,000 images from the training set** and **1000 images from the test set** and re-generated the corresponding feedback with GPT-4o. As shown in Table A, although the choice of feedback generator does affect retrieval performance, the gap remains relatively small, and the performance trend across turns is consistent, indicating that the distilled InternVL 2.5-8B provides comparable feedback quality of GPT-4o and is suitable for scalable dataset construction.
>
> **Table A: Retrieval performance using different feedback generators.**
>
> | Generator                   | R-1   | R-5   | R-10  | mAP   |
> | --------------------------- | ----- | ----- | ----- | ----- |
> | GPT-4o                      | 83.76 | 97.34 | 98.92 | 82.36 |
> | InternVL 2.5-8B (distilled) | 83.15 | 96.95 | 98.70 | 81.84 |
>
> **(2) Manual Curation Statistics:** During the dataset construction, we employed 22 human annotators to review the generated feedback. They were instructed to correct erroneous descriptions and complete missing details as shown in Appendix D.2. The curation statistics are summarized in Table B. These statistics show that although the generated feedback is usable in most cases, manual curation is essential for correcting factual errors and enriching incomplete responses.
>
> **Table B: Statistics of the manual curation.**
>
> | Curation            | Percentage |
> | ------------------- | ---------- |
> | Correct Feedback    | 74.22%     |
> | Generation Errors   | 14.35%     |
> | Incomplete Feedback | 11.43%     |
>
>
>
> ---
>
> **[W2] No real user evaluation**
>
> To validate practical applicability, we conducted a small-scale real-user study. We invited **10 human evaluators** to interact with our system on **100 target identities**. As shown in Table C, compared with synthetic feedback, although human feedback is naturally noisier and more diverse, MNEMO still achieves substantial improvements over turns, showing that MNEMO is robust to realistic user feedback.
>
> **Table C: Comparison of synthetic and human feedback (R-1 reported).**
>
> | Feedback Source | Turn 0 | Turn 1 | Turn 2 |
> | --------------- | ------ | ------ | ------ |
> | Synthetic       | 78.09  | 91.77  | 92.26  |
> | Human           | 78.09  | 89.56  | 90.14  |
>
>
>
> ---
>
> **[W3] Limited backbone scaling analysis**
>
> Thank you for this constructive suggestion. We additionally extended ablation study to **InternVL 2.5-4B**. As shown in Table D, we observe the same trend on both backbone scales, consistent with our analysis in **Section 5.3.1**:
>
> - **ATE** brings substantial efficiency gains by encoding each dialogue turn separately, avoiding the long-context overhead introduced by direct dialogue concatenation;
>
> - **DMA** boosts the retrieval performance by aggregating contextual information across dialogue turns, while preserving most of the efficiency gains brought by ATE.
>
> Overall, the combination of ATE and DMA achieves the best performance on both 1B and 4B, showing that the functional roles of ATE and DMA remain consistent on larger backbones.
>
>  **Table D: Scaling analysis of ATE and DMA.**
>
> | Backbone                  | R-1 $\uparrow$ | mAP $\uparrow$ | Training Time (ms) $\downarrow$ | Inference Time (ms) $\downarrow$ |
> | ------------------------- | -------------- | -------------- | ------------------------------- | -------------------------------- |
> | InternVL 2.5-1B           | 91.51          | 78.26          | 661.78                          | 63.76                            |
> | InternVL 2.5-1B +ATE      | 88.90          | 76.99          | 176.25                          | 22.77                            |
> | InternVL 2.5-1B +ATE +DMA | 92.26          | 80.31          | 189.24                          | 24.07                            |
> | InternVL 2.5-4B           | 92.74          | 79.53          | 4036.86                         | 432.93                           |
> | InternVL 2.5-4B +ATE      | 90.31          | 78.28          | 1029.30                         | 139.58                           |
> | InternVL 2.5-4B +ATE +DMA | 93.43          | 81.44          | 1177.07                         | 156.21                           |
>
>
>
> ---
>
> We will include these results in the revised manuscript. We hope these additional experiments and clarifications help address your concerns. We would be grateful if you would take these new results into consideration in your final assessment.

---

> > ### Author Rebuttal · Reviewer_SDm5 · 2026-04-03
> >
> > My concerns have been adequately addressed, and I would like to maintain my positive rating.

---

> > > ### Author Response · Authors · 2026-04-04
> > >
> > > We sincerely appreciate your recognition of our paper. We are truly encouraged to hear that your concerns have been adequately addressed, and we are grateful for your positive rating of 4. Thank you for your time, thoughtful feedback, and continued support.

---

### Decision · Program_Chairs · 2026-04-30

**Decision:**

Accept (regular)

**Comment:**

This paper introduces a novel paradigm for Multimodal Interactive Person Retrieval (MInterPR), supported by a new conversational dataset (MInterPEDES) and a memory-enhanced retrieval framework (MNEMO). Reviewers universally praised the work for its well-motivated problem setting, clear presentation, and technically sound architecture that effectively balances computational efficiency with contextual modeling through its Atomic Turn Encoding and Dialogue Memory Aggregation modules. During the initial review phase, reviewers raised valid concerns regarding the dataset's heavy reliance on synthetic, offline MLLM-generated feedback, the lack of real-user interaction validation, and missing comparisons with recent benchmarks. In a thorough rebuttal, the authors successfully mitigated these issues by conducting a small-scale real-user study, evaluating memory-based feedback to simulate realistic human uncertainty, expanding dataset comparisons, and providing additional baseline and scaling analyses. While some reviewers noted that a more extensive online user study would further solidify the benchmark's external validity, they unanimously agreed that the rebuttal adequately resolved their core concerns. Given the originality of the interactive retrieval task, the solid empirical performance, and the positive final consensus across all reviewers, this paper is recommended for acceptance.